# Protein NirP1 regulates nitrite reductase and nitrite excretion in cyanobacteria

Alexander Kraus[1], Philipp Spät [2], Stefan Timm [3], Amy Wilson[1], Rhena Schumann [4], Martin Hagemann [3], Boris Maček [2] & Wolfgang R. Hess [1] ✉

When the supply of inorganic carbon is limiting, photosynthetic cyanobacteria excrete nitrite, a toxic intermediate in the ammonia assimilation pathway from nitrate. It has been hypothesized that the excreted nitrite represents excess nitrogen that cannot be further assimilated due to the missing carbon, but the underlying molecular mechanisms are unclear. Here, we identified a protein that interacts with nitrite reductase, regulates nitrogen metabolism and promotes nitrite excretion. The protein, which we named NirP1, is encoded by an unannotated gene that is upregulated under low carbon conditions and controlled by transcription factor NtcA, a central regulator of nitrogen homeostasis. Ectopic overexpression of *nirP1* in *Synechocystis* sp. PCC 6803 resulted in a chlorotic phenotype, delayed growth, severe changes in amino acid pools, and nitrite excretion. Coimmunoprecipitation experiments indicated that NirP1 interacts with nitrite reductase, a central enzyme in the assimilation of ammonia from nitrate/nitrite. Our results reveal that NirP1 is widely conserved in cyanobacteria and plays a crucial role in the coordination of C/N primary metabolism by targeting nitrite reductase.

Some cyanobacteria, such as marine *Prochlorococcus* and *Synechococcus*, are of paramount importance as primary producers and for the global biogeochemical carbon (C) and nitrogen (N) cycles[1–3]. Other cyanobacteria, such as *Synechococcus* sp. PCC 7942 (*Synechococcus* 7942) or *Synechocystis* sp. PCC 6803 (*Synechocystis* 6803), have become model species for the $CO_2$-neutral production of chemical feedstock and biotechnological processes driven by photosynthesis[4,5]. The two most relevant nutrients for cyanobacteria are inorganic carbon ($C_i$) and N. Most cyanobacteria assimilate combined inorganic N forms, such as ammonium ($NH_4^+$), nitrate ($NO_3^-$), nitrite ($NO_2^-$), and urea; in addition, diazotrophic species can utilize $N_2$ gas.

Surprisingly, several different cyanobacteria excrete nitrite under photoautotrophic growth conditions. Certain isolates of the marine cyanobacterium *Prochlorococcus* were reported to release up to 30% of their N uptake as extracellular nitrite when grown on nitrate[6].

The released nitrite is cross-fed by other microbes that cannot consume nitrate or produce nitrite. Moreover, it was demonstrated that cooccurring microbes could be beneficial for *Prochlorococcus*[7,8]. In *Synechococcus* 7942, as well, approximately 30% of the produced nitrite were excreted in shifts to low $C_i$ (LC) conditions, while no nitrite was excreted under high $C_i$ (HC) conditions[9].

The coordination of C and N metabolism is regulated by several transcription factors and other regulatory proteins. An acute N scarcity is sensed by two central regulators of N assimilation in cyanobacteria, $P_{II}$ and NtcA, by binding the key metabolite 2-oxoglutarate (2-OG)[10–14]. 2-OG connects C and N metabolism because it represents the main precursor for ammonia assimilation through the glutamine synthetase/glutamine oxoglutarate aminotransferase (GS/GOGAT) cycle in cyanobacteria and plants. Initially, GS incorporates ammonia into glutamate, thereby producing glutamine. The amino group from

[1]Genetics and Experimental Bioinformatics, Faculty of Biology, Freiburg University, D-79104 Freiburg, Germany. [2]Department of Quantitative Proteomics, Interfaculty Institute for Cell Biology, University of Tübingen, D-72076 Tübingen, Germany. [3]Plant Physiology Department, Institute of Biosciences, University of Rostock, D-18059 Rostock, Germany. [4]Biological Station Zingst, University of Rostock, D-18374 Zingst, Germany. ✉e-mail: wolfgang.hess@biologie.uni-freiburg.de

glutamine is then transferred onto 2-OG by GOGAT, yielding two glutamate molecules. As a consequence, the intracellular level of 2-OG starts to increase once these reactions slow down due to insufficient N supply; thus, 2-OG is an excellent indicator of the N status[15]. Binding of 2-OG stimulates the activity of NtcA, the main transcriptional regulator of genes encoding proteins involved in N assimilation[10,11]. Depending on the 2-OG level, the $P_{II}$ interaction protein X (PipX) switches from binding to $P_{II}$ to interacting with NtcA, further enhancing the binding affinity of this complex to target promoters[12,14]. In *Synechocystis* 6803, NtcA directly activates 51 genes, including transporters for N sources and GS/GOGAT, and represses 28 other genes after 4 h of N starvation[16].

In addition to transcription factors, multiple different proteins have been identified that interact with these factors or the involved enzymes and govern their functions directly. Archetypical examples for these proteins are the inhibitory factors IF7 and IF17, as their binding to GS leads to the inactivation of the enzyme when N supply exceeds demand[17]. The $P_{II}$-interacting regulator of arginine synthesis (PirA) regulates the flux into the ornithine-ammonia cycle and therefore arginine synthesis[18,19], while the $P_{II}$-interacting regulator of C metabolism (PirC) functions as an inhibitor of 2,3-phosphoglycerate–independent phosphoglycerate mutase (PGAM), the enzyme that shifts newly fixed $CO_2$ toward lower glycolysis[20]. PirC becomes sequestered by $P_{II}$ at low 2-OG levels and released for PGAM inhibition at high 2-OG levels and therefore connects the control of C and N assimilatory pathways in a particular way[20].

We have previously analyzed the primary transcriptomes of *Synechocystis* 6803 and the closely related strain *Synechocystis* sp. PCC 6714[21–23]. Based on these analyses, many more genes encoding unknown small, potentially regulatory proteins were computationally predicted. Experiments using a 3xFLAG epitope tag fused in-frame to the 3′ ends of the respective reading frames validated five of these to encode small proteins[24]. Subsequent experimental analyses using coimmunoprecipitation (co-IP) demonstrated that one of these small proteins, NblD, facilitates the degradation of phycobilisomes under N starvation conditions[25], while another one, called AtpΘ, is an inhibitor of the ATP synthase back reaction under low energy conditions[26]. Collectively, these findings suggest that cyanobacteria use a broad collection of small protein modulators to regulate metabolism according to external clues.

Here, we report a small, regulatory protein involved in the control of N assimilation in cyanobacteria in an unexpected way. This protein is 81 amino acids long in *Synechocystis* 6803, and its expression is upregulated following shifts from HC to LC conditions and downregulated upon N downshifts. Co-IP experiments identified the enzyme nitrite reductase (NiR) as the primary target of this protein. Following this lead further, we characterize the protein as a regulatory factor in the primary assimilation of N from nitrate/nitrite. Based on the observed phenotype, we named the regulatory protein NirP1, for nitrite reductase interacting protein 1.

## Results

### The ncr1071 transcript exhibits coding potential

The NtcA regulon in *Synechocystis* 6803 was reported to comprise 79 genes, including eight non-coding RNAs (ncRNAs)[16]. One of these ncRNAs, with the transcript ID ncr1071, was strongly downregulated upon N step-down[16]. This transcript extends from position 2215954 to 2217039 on the forward strand of the *Synechocystis* 6803 chromosome (GenBank accession no. NC_000911) and was assigned to the transcriptional unit (TU) 2296[21]. When 10 different cultivation conditions were compared, TU2296 was found to be maximally transcribed under LC, while it was strongly repressed under N starvation conditions[21]. TU2296 overlaps the coding sequence of the *sll1864* gene encoding a potential chloride channel protein on the reverse strand by more than 500 nt (Fig. 1a).

Upon closer inspection, we discovered that the first segment of this transcript exhibits potential to encode an 81-residue protein with a calculated IEP of 4.59 and a molecular mass of 9.18 kDa. Evidence for the possible existence of this protein was provided independently by mass spectrometry data from a deep proteogenomics dataset[27].

Based on the results below, we named this protein NirP1, for nitrite reductase interacting protein 1. Database searches identified 485 potential homologs of *nirP1* (Supplementary Dataset 1) in species belonging to all morphological subsections of cyanobacteria[28], including the well-studied model *Synechococcus* 7942, heterocyst-forming diazotrophic species and Section V strains, such as *Fischerella*. Homologous genes were detected in marine species with deviating pigmentation, such as *Acaryochloris marina* and *Halomicronema hongdechloris*, but not in *Prochlorococcus* and not in the majority of marine picoplanktonic *Synechococcus* strains. Homologs were also not detected in the genomes of diatom-associated symbionts *Calothrix rhizosoleniae* and *Richelia intracellularis* nor in the endosymbiont chromatophore genomes of photosynthetic *Paulinella* species or the UCYN-A *Candidatus* Atelocyanobacterium thalassa endosymbionts, which can perform $N_2$ fixation but lack photosystem II and phycobilisomes[29,30]. These data suggest that homologs of *nirP1* were advantageous but not essential for free-living cyanobacteria with very different physiologies.

The vast majority of the 485 putative homologs identified in this work (Supplementary Dataset 1) are between 76 and 86 aa in length. The alignment of selected NirP1 homologs highlights the conservation in the last ~50 aa and the presence of a centrally located, absolutely conserved single cysteine and three tryptophan residues (Fig. 1b).

To verify the inducibility of *nirP1* transcription by shifts to LC conditions, cultures of *Synechocystis* 6803 precultivated in medium supplemented with 10 mM NaHCO₃ (high $C_i$, HC) were centrifuged and resuspended in medium lacking any source of $C_i$. In Northern blot analysis, samples taken over 24 h showed a single signal of approximately 580 nt (Fig. 1c) matching the length from the transcription start site (TSS) to the steep decline in the number of reads approximately 80 nt after the *nirP1* stop codon (Fig. 1a). A strong signal was obtained 1 h after the shift, with a slightly decreasing intensity at the later time points and increasing again at the 24 h sampling point. At a weaker intensity, the *nirP1* mRNA was also detectable in the sample from high $C_i$ (HC)-grown cells.

To directly validate the presence of NirP1, strains expressing NirP1 −3xFLAG fusions were constructed (Supplementary Fig. S1). Protein extracts were prepared and analyzed by SDS–polyacrylamide gel electrophoresis (SDS–PAGE) and Western blotting using anti-FLAG antiserum. Two signals of slightly different molecular masses were detected. The calculated molecular mass of ~12 kDa for the prominent signal in the samples from HC-grown cultures was consistent with the sum of the calculated molecular masses of 9.18 kDa for the monomeric NirP1 and 2.86 kDa for the 3xFLAG tag (Fig. 1d). After the shift to LC conditions, a band migrating with a ~1.5 kDa larger molecular mass became more intense, possibly indicating a post-translational modification. No FLAG-tagged NirP1 was detected in the wild-type strain used as a control. From these experiments, we conclude that TU2296 encodes the small protein NirP1 and that *nirP1* transcription responds to shifts in $C_i$ supply.

To generate a model for the NirP1 structure, we used AlphaFold[31,32], which modeled NirP1 as a monomer and predicted four beta folds in the most conserved part of the protein. No structure was predicted for the N-terminal section of the protein, consistent with a lack of sequence conservation (Fig. 1b). Two cysteine residues (Cys27 and Cys81) in *Synechocystis* 6803 NirP1 were predicted to be in close proximity to each other, potentially forming a disulfide bond (Fig. 1e), whereas a third, highly conserved Cys55 was found between two beta strands in the folded part, exposed to the outside. However, no functional domains could be predicted based on the primary sequence or modeled structure.

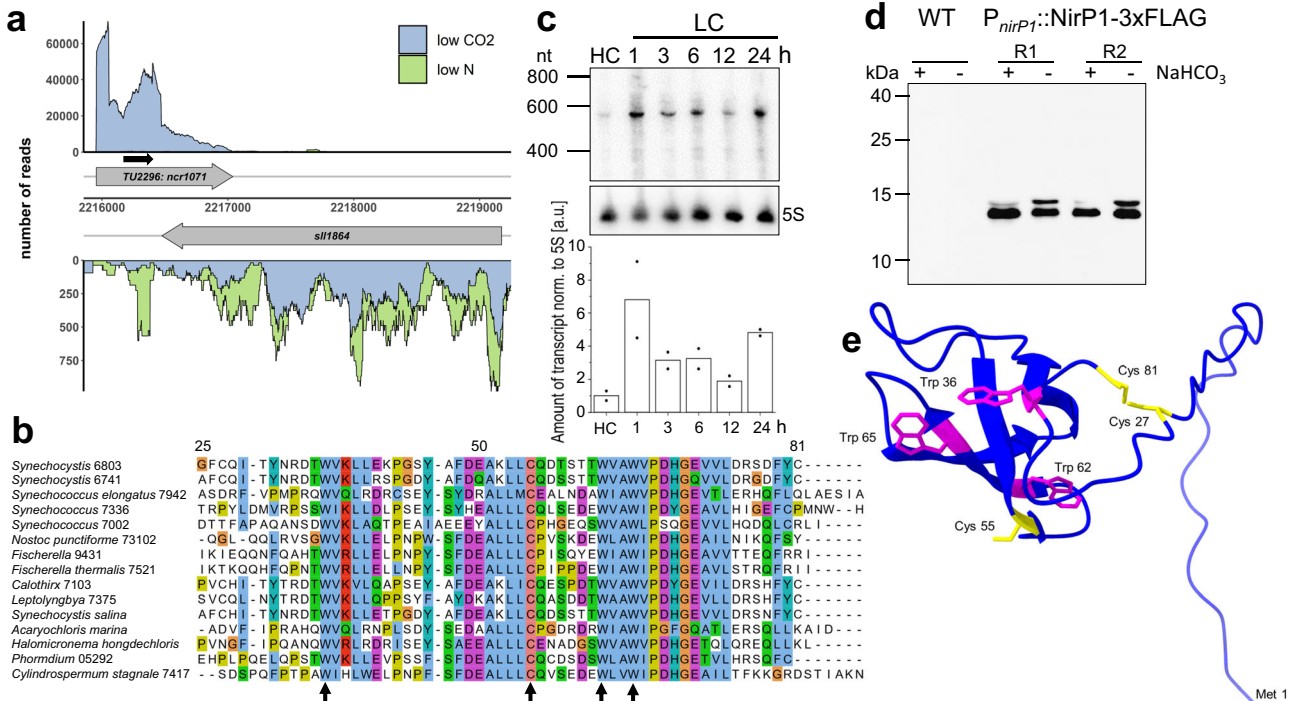

**Fig. 1 | Genomic location and expression of *nirP1* in *Synechocystis* 6803. a** The transcriptional unit TU2296 encompassing the coding sequence of *nirP1* (*ncr1071*) overlaps gene *sll1864* annotated as encoding a chloride ion channel protein. Plot at low $CO_2$ (blue) versus N starvation (green). Transcriptional units (TUs) are indicated according to the previous annotation of the transcriptome and genome-wide mapping of transcriptional start sites[21]. The potential open reading frame of *ncr1071* coding for the NirP1 protein is indicated by a black arrow. **b** Alignment of selected potential NirP1 homologs from cyanobacteria belonging to four morphological subsections[28]. The last and most conserved part of *Synechocystis* 6803 NirP1 from position 25 to the end is shown. The cysteine and tryptophan residues conserved in all 485 potential homologs (Supplementary Dataset 1) are marked by arrows. The alignment was generated using ClustalW and visualized by Jalview. **c** Time course of *nirp1* mRNA accumulation in *Synechocystis* 6803 after cells pre-cultivated in medium supplemented with 10 mM NaHCO3 (HC) for 3 h were shifted to medium without a source of $C_i$ (LC). The *nirp1* transcript was detected by a $^{32}$P-labeled single-stranded RNA probe. The membrane was rehybridized to a 5S rRNA probe as a loading control. The RiboRuler Low Range RNA ladder (Thermo

Fisher Scientific) was used as the molecular mass standard. The relative amount of *nirp1* transcript normalized to the 5S rRNA is shown in the bar chart. The HC condition was set to 1, and all other signals were normalized to the HC condition. Two independent biological replicates were used and averaged. **d** NirP1 was detected in the presence or absence of $C_i$ by Western blotting using anti-FLAG antiserum against tagged NirP1 under the control of its native promoter in two biological replicates (R1 and R2). The wild type was used as a control. Cultures were grown in medium supplemented with 10 mM NaHCO3 (+), washed, and cultivated in carbonate-free medium (-) for 24 h. Prestained PageRuler™ (Thermo Fisher Scientific) was used as a molecular mass marker. The blot has been performed in two independent experiments (*n* = 2). **e** NirP1 structure predicted by AlphaFold[31,32] for the complete *Synechocystis* NirP1 protein, indicating the presence of four beta folds in the most conserved part. The four totally conserved amino acids as well as Cys27 and Cys81 are highlighted, tryptophan is shown in magenta, and cysteine in yellow. The Met1 residue is also indicated for orientation. Source data are provided as a Source Data file.

## Two transcription factors are involved in the regulation of *nirP1* expression

Based on literature reports[16], we speculated that *nirP1* expression could be negatively regulated by NtcA binding to a conserved binding site over the TSS. Moreover, we observed a tandem repeat element located further upstream consisting of two octameric repeats TTTGT(T/C)AA separated by a dinucleotide. In promoter sequence analyses of 456 *nirP1* homologs (Supplementary Dataset 2), the NtcA-binding motif was present and conserved in 435 cases. The tandem repeat was found in 135 homologs (Supplementary Dataset 2; Supplementary Fig. S2). To investigate the possible regulation through these sequence elements, the promoter of *nirP1* (P*nirP1*; from −70 to +30, TSS at +1) was engineered in a transcriptional fusion to drive the *luxAB* reporter genes. We used the native P*nirP1* sequence or promoter variants with mutated, likely relevant nucleotides either in the NtcA-binding site (P_NtcA-Mut) or in the upstream promoter element (P_Repeat-Mut; Fig. 2a).

In strains P*nirP1* and P_NtcA-Mut, transcription was activated 1 h after a shift to medium lacking $C_i$, followed by a decrease and a light increase again at the later time points (Fig. 2b, Supplementary Fig. S3). The increase at 24 h was not as pronounced as in the Northern blot (Fig. 1c), pointing at the possible involvement of post-transcriptional regulation. This activation was significantly higher in the P_NtcA-Mut strains with

the mutated NtcA-binding site. This effect was even more pronounced after a shift to N starvation conditions (Fig. 2c; Supplementary Dataset 3). These results indicated the repressor function of NtcA for *nirP1*, consistent with previous ChIP-seq[16] and transcriptome data[21]. However, repression was still observed 6 h after the shift to N starvation, even when the binding motif was mutated rendering NtcA binding impossible, indicating the presence of additional regulatory mechanisms. It is possible that a second transcription factor is involved in the repression of *nirP1* during N starvation.

If the tandem repeat was mutated (P_Repeat-Mut), a signal close to baseline was detected in all tested conditions (Fig. 2b–d, Supplementary Dataset 3; Supplementary Fig. S3). We conclude that this motif is essential for the transcription of *nirP1*, consistent with its position centered at −51.5 from the TSS at +1 (Fig. 2a), a region where frequently transcription factors bind to activate transcription.

Finally, transcriptional regulation was tested in response to the addition of ammonium (Fig. 2d). The signal increased ~threefold after 1 h and then decreased, similar to the results obtained after the shift in LC conditions (Fig. 2b). These data agree with the results of a recent transcriptome analysis in which the *ncr1071* transcript was upregulated ~threefold during acclimation to ammonium stress[33], indicating that the C/N balance is pivotal for the transcription of *nirP1*.

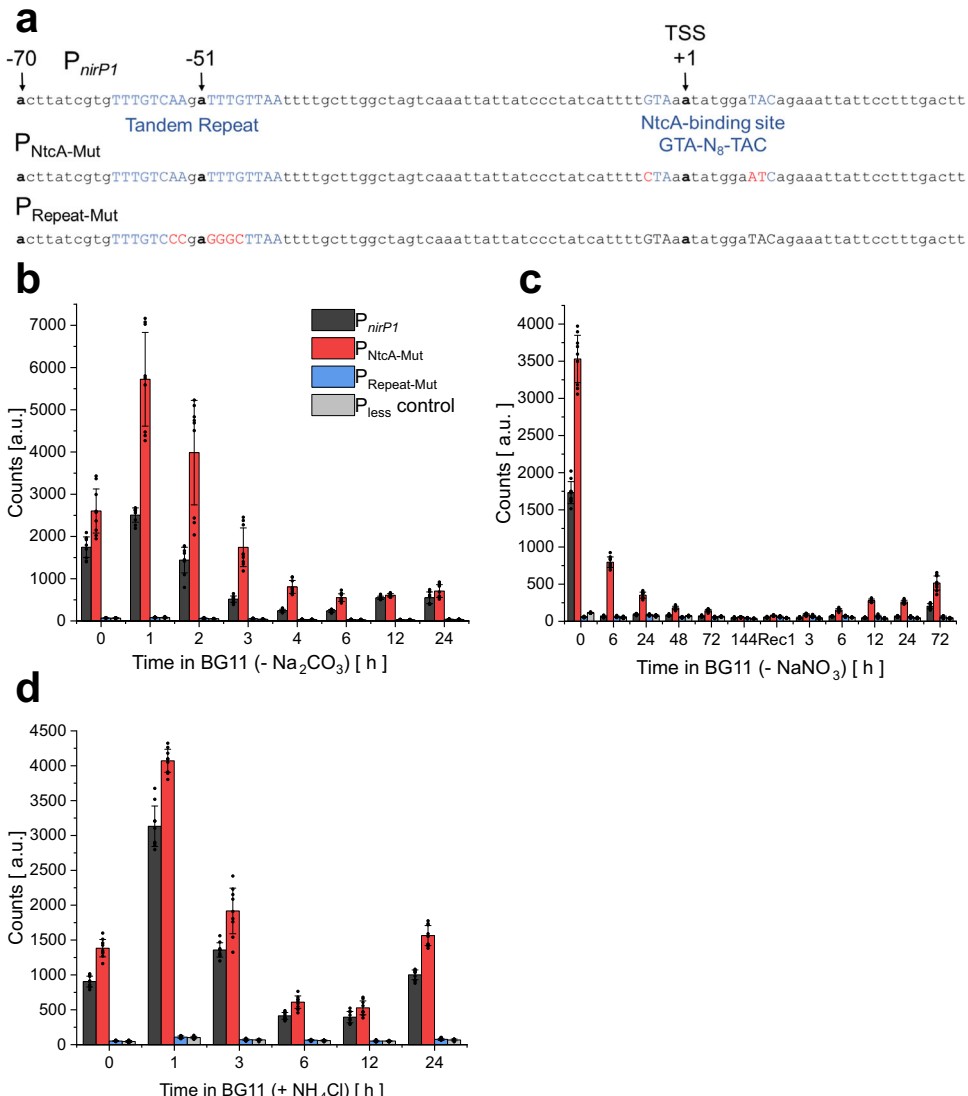

**Fig. 2 | NirP1 expression is mediated through a $C_i$-sensitive promoter and the transcription factor NtcA. a** Top: Native $P_{nirP1}$ promoter sequence from position −70 to +30 (TSS at +1) used in a bioluminescence reporter strain harboring a $P_{nirP1}$-*luxAB* transcriptional fusion. Functional promoter elements are colored blue. The previously determined[21] transcription start site (TSS) is indicated. Middle row: Mutated nucleotides in the NtcA-binding site yielding promoter $P_{NtcA-Mut}$ are colored red. Bottom: Mutated nucleotides in the repeat motif yielding promoter $P_{Repeat-Mut}$ are colored red. Both mutated promoters were fused to *luxAB* and introduced into a neutral site in parallel with the $P_{nirP1}$-*luxAB* construct. **b** Bioluminescence of the *Synechocystis* 6803 $P_{nirP1}$-*luxAB*, $P_{NtcA-Mut}$-*luxAB* reporter, and $P_{Repeat-Mut}$-*luxAB* strains and a promoter-less ($P_{less}$) negative control. Cells were grown under HC conditions (BG11 supplemented with 10 mM NaHCO$_3$) for 2 h and transferred to BG11 medium without a $CO_2$ source to induce LC conditions for 24 h. **c** The strains were cultivated in BG11 (-N), and the chlorotic cultures were transferred back to standard BG11 (17.6 mM NaNO$_3$) medium to start the recovery process seven days after N starvation was initiated. **d** Strains were cultivated in standard BG11 and then 10 mM NH$_4$Cl was added for 24 h. Bioluminescence data are presented as the means ± SDs of 3 independent measurements with three biological replicates each ($n = 9$). Significance was calculated with a two-tailed *t*-test with unequal variance (Welch's *t*-test; *$P < 0.05$; **$P < 0.01$; ***$P < 0.001$) between the strains at corresponding time points (details of statistical analysis Supplementary Fig. S3 and in Supplementary Dataset 3). Source data are provided as a Source Data file.

Collectively, these results demonstrated the complex transcriptional regulation of *nirP1*. We validated the transcriptional regulation of *nirP1* by NtcA and obtained new insight into the regulation of *nirP1* in shifts to N deprivation and with different C/N ratios. Due to the observed strong regulation, we hypothesized that NirP1 might be a regulator and is likely involved in the acclimation to changing $C_i$ and N supply.

### Deletion and ectopic expression of NirP1 leads to phenotypical changes

To test the potential functional importance of NirP1, we constructed the following strains: a gene deletion strain Δ*nirP1*; two overexpression strains in which the coding sequence of *nirP1* was fused to a C-terminal 3xFLAG tag under control of the copper-inducible $P_{petE}$ promoter on a plasmid pVZ322 introduced in a wild type background (strain NirP1oex) and in the Δ*nirP1* background (strain Δ*nirP1*::oex); and a complementation strain with the same plasmid as in the overexpression strains, except that *nirP1* was controlled by its native promoter (Δ*nirP1*::*nirP1*).

Under nutrient-replete conditions, the Δ*nirP1* deletion mutant showed linear growth similar to the wild type. In contrast, both strains expressing *nirP1* under control of the $P_{petE}$ promoter showed severely reduced growth in liquid medium, irrespective of whether the plasmid was in the wild type or the Δ*nirP1* background (Fig. 3a).

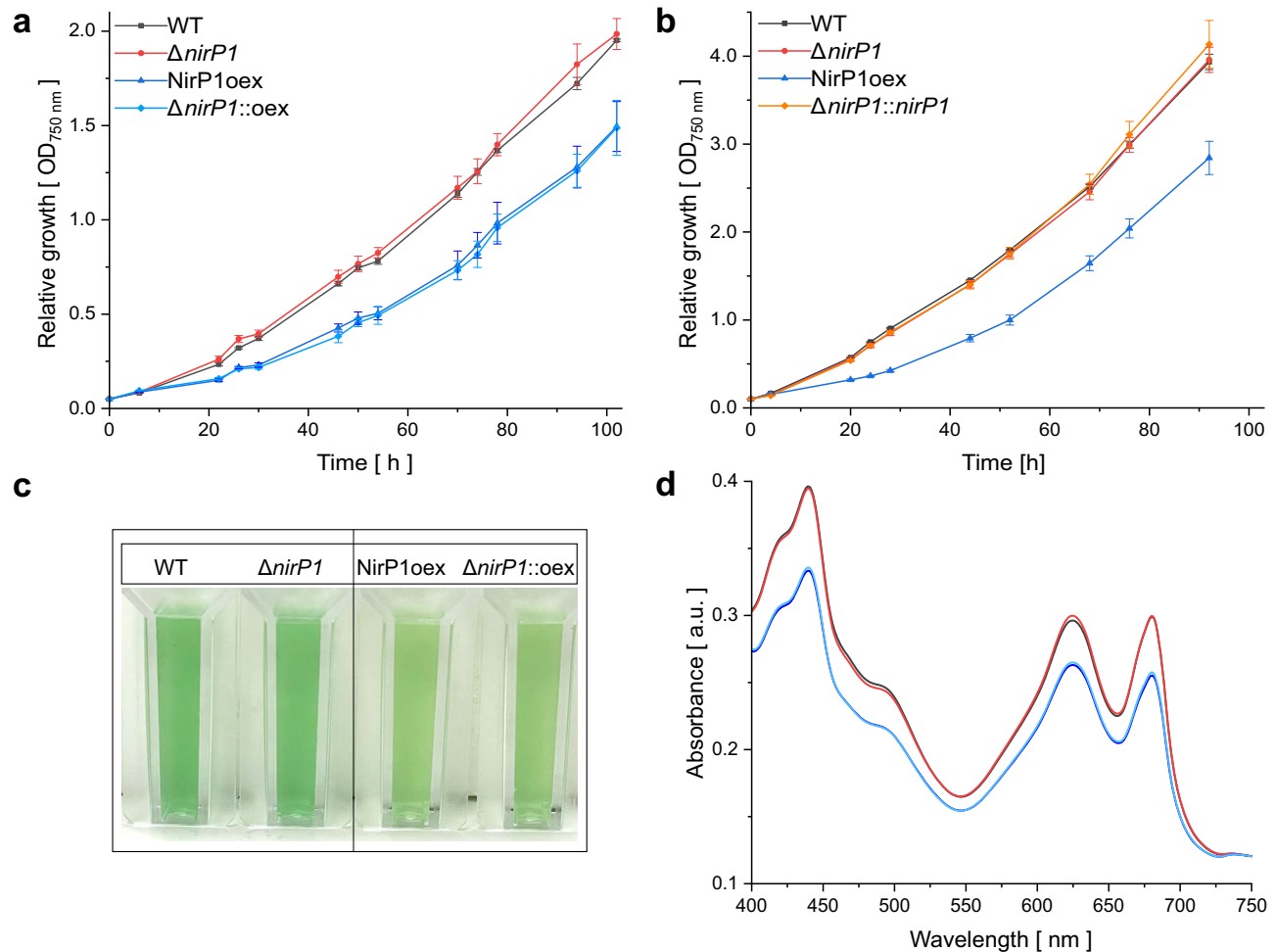

**Fig. 3 | Phenotypical differences between *nirP1* mutant strains and the wild type in BG11 medium. a** Growth of wild type (black), the Δ*nirP1* deletion mutant (red), and the strains expressing *nirP1* under control of the P$_{petE}$ promoter in the wild-type background (dark blue, NirP1oex) and Δ*nirP1* background (light blue, Δ*nirP1*::oex). Data points represent the mean ± SD of 3 biological replicates (*n* = 3; details with growth rates and doubling time are provided in Supplementary Dataset 4). **b** Growth in liquid medium after shifting to LC conditions. Strains were cultivated for 2 h in medium containing 10 mM NaHCO$_3$, collected, and transferred to medium without a C$_i$ source. The complementation strain (orange, Δ*nirP1*::*nirP1*) expressed *nirP1* under the control of its native P$_{nirP1}$ promoter in the Δ*nirP1* background. Other strains as in (**a**); *n* = 3. **c** Pigmentation phenotype of wild-type and *nirP1* overexpressors in the presence of Cu$_2$SO$_4$ expressing *nirP1* for 24 h. All cultures were set to an OD of ~0.8. **d** Room temperature absorption spectra for wild-type and *nirP1* mutants expressing *nirP1* for 24 h from the P$_{petE}$ promoter in the presence of Cu$_2$SO$_4$, normalized to wild-type OD$_{750}$. Same strains and colors as in (**a**). Spectra were recorded from three biological replicates and averaged (*n* = 3). Further phenotypical differences in Supplementary Figs. S4 and S5.

The reduced growth observed at high *nirP1* expression compared to wild type and Δ*nirP1* was also visible under LC, while the strain grew normally if *nirP1* was controlled by its native P$_{nirP1}$ promoter (Fig. 3b). Therefore, we concluded that the high expression of *nirP1* from the P$_{petE}$ promoter led to reduced growth, in particular, to a prolonged lag phase (Supplementary Dataset 4). In addition, we noticed a pigmentation change, i.e., the cultures became chlorotic if *nirP1* was overexpressed from the P$_{petE}$ promoter (Fig. 3c). This effect was strictly dependent on the *nirP1* expression level because cells of the non-induced mutant showed the same pigmentation as the wild type in the absence of added Cu$_2$SO$_4$. (Supplementary Fig. S4a). This phenotype was verified in room temperature absorption spectra. Compared to the spectra of the wild type and deletion mutant, the phycobilisome peak at approximately 625 nm as well as the chlorophyll peaks at 440 and 680 nm were decreased in the two strains that expressed *nirP1* from P$_{petE}$ after addition of Cu$_2$SO$_4$ (Fig. 3d, Supplementary Dataset 4).

To extend these results, drop dilution assays were performed to compare growth on plates. Again, the strain that overexpressed *nirP1* from the P$_{petE}$ promoter showed reduced vitality compared to the wild

type and Δ*nirP1*, irrespective of the addition of 10 mM NaHCO$_3$ (Supplementary Fig. S4b). In conditions where nitrate was replaced by ammonium as the only N source, the phenotype was completely abolished (Supplementary Fig. S5). To summarize, the phenotypical effects due to ectopic NirP1 expression suggested that the amount of NirP1 played a critical role in the cell in presence of nitrate, while its absence did not lead to major phenotypic effects under the tested conditions.

## Metabolic changes in the GS/GOGAT and ammonia-ornithine cycle

Upon *nirP1* overexpression, the chlorotic phenotype (Fig. 3c, d) resembled the pigmentation changes characteristic of N-starved cells. Together with the observed expression control by NtcA (Fig. 2), this suggested a primary relationship between NirP1 and processes related to the assimilation of N, albeit the regulatory effect of LC.

Therefore, we tested whether the ectopic expression of NirP1 influences amino acid levels in the cell. Metabolites and amino acids were extracted in a time course experiment from wild type, Δ*nirP1*, and NirP1oex following a shift to LC. The total amount of soluble amino

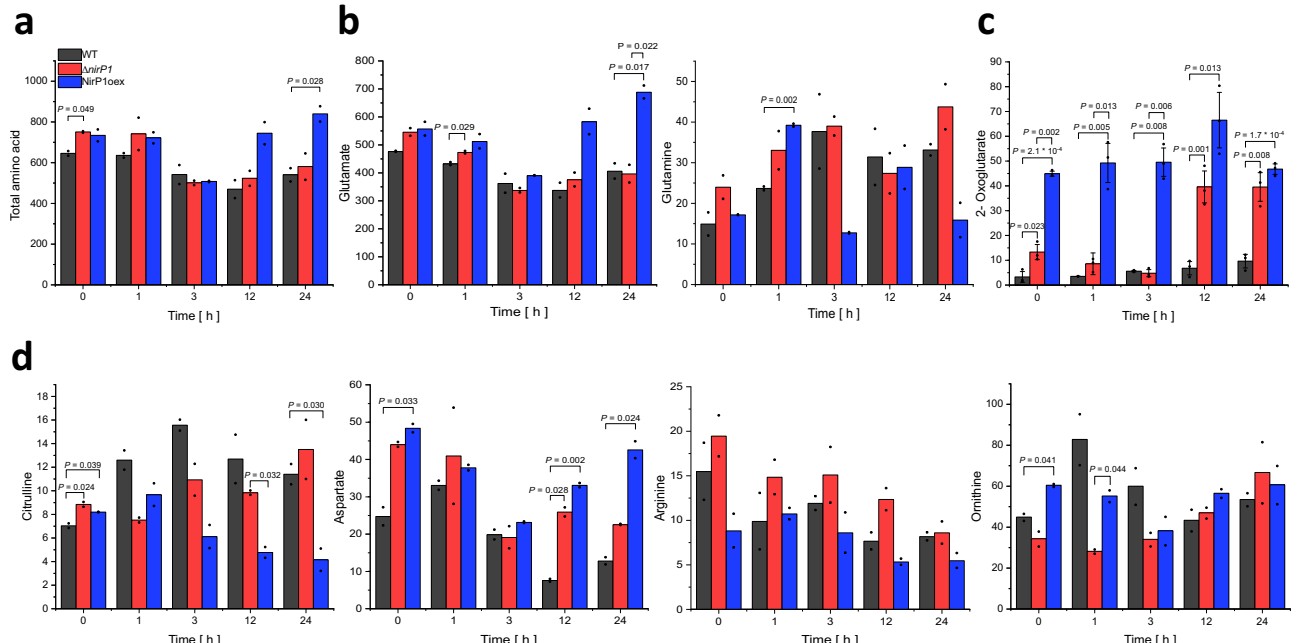

**Fig. 4 | Differences in the accumulation of key intermediates in C/N primary metabolism due to the availability of NirP1 during shifts to LC. a** Total amino acid content. **b** Glutamate and glutamine concentrations. **c** 2-oxoglutarate concentrations. **d** Concentrations of ornithine-ammonium cycle intermediates and of aspartate. Metabolite content was measured for Δ*nirP1*, NirP1oex, and wild type (WT). All concentrations are given in ng * mL$^{-1}$ * OD750 nm$^{-1}$. Two biological replicates (*n* = 2) were used for all strains and metabolites (except in **c** where the means ± SD of three biological replicates are shown, *n* = 3) and averaged (details and raw data in Supplementary Dataset 5). Significance was calculated with the two-tailed *t*-test with unequal variance (Welch's *t*-test; *only *P* ≤ 0.05 are shown) for each strain and between the strains at corresponding time points. The measurements were independently repeated in triplicates (*n* = 3) and the results confirmed (details in Supplementary Dataset 6).

acids was significantly higher in NirP1oex at 24 h after transfer to LC than in the other two strains (Fig. 4a, full data in Supplementary Dataset 5). However, a closer look at the data revealed that the higher amino acid levels in NirP1oex were mainly caused by the accumulation of a single amino acid, glutamate (Fig. 4b). The glutamate level in *Synechocystis* 6803 is normally approximately 10 times higher than that of any other amino acid. Therefore, the increased accumulation of glutamate alone was responsible for the measured higher total amino acid amount in NirP1oex. However, for the direct product of glutamate amination by GS, glutamine, we measured lower levels in NirP1oex at 3 and 24 h after the shift to LC (Fig. 4b). This result is consistent with previous analyses in which a decrease in glutamine content was observed in LC shifts[34]. In addition, we found that the concentrations of citrulline and arginine, which are produced from glutamine via the ornithine-ammonium cycle (OAC), were lowered in NirP1oex at 3, 12, and 24 h after the induction of expression (Fig. 4d). In contrast, the aspartate levels were increased, consistent with its synthesis from glutamate by glutamate aspartate aminotransferase. These tendencies were confirmed in a repeat experiment (Supplementary Dataset 6). Furthermore, the concentration of 2-oxoglutarate (2-OG) was significantly higher in NirPoex than in the other strains (Fig. 4c). This finding indicates an impaired regulation of the GS/GOGAT cycle and a reduced N assimilation caused by high NirP1 and is consistent with effect the of pigmentation (Fig. 3c) and glutamine and glutamate concentration (Fig. 4b). In the wild type and Δ*nirP1* deletion mutant, 2-OG levels increased after 12 and 24 h, which was more pronounced in the deletion mutant pointing to a further regulation when NirP1 is absent (Fig. 4c). We conclude that the upregulation of NirP1 caused several changes in amino acid levels related to the synchronization between C and N metabolism. As the OAC is involved in N storage and remobilization[35], these results support that NirP1 plays a role as a regulator in N metabolism.

## NirP1 interacts with ferredoxin-nitrite reductase

Coimmunoprecipitation (co-IP) was successfully used in the past to identify the interacting partners of *Synechocystis* 6803 small proteins NblD, AtpΘ, and SliP4, thereby providing essential hints for their specific functions[25,26,36]. Therefore, we used a strain-expressing epitope-tagged NirP1 to obtain insight into the molecular mechanism that involves NirP1. After 24 h of induction, total cell extracts of three replicates of NirP1oex were analyzed by SDS–PAGE and Western blotting using anti-FLAG antisera. The size of the NirP1-3xFLAG signal in the Western blot matched the sum of the calculated molecular masses of 9.81 kDa for NirP1 and 2.86 for the 3xFLAG tag. No signal was detected in the wild type used as a negative control (Fig. 5a).

NirP1-3xFLAG and potential bound interaction partners were coimmunoprecipitated from the lysate using anti-FLAG resin (Fig. 5b). Elution fractions of the protein pull-down showed NirP1 in Coomassie-stained protein gels and an additional band of higher molecular mass in both replicates. Because signals were obtained in the Western blot at these two and a third position, this result pointed at potential interaction partners of NirP1 that were copurified with the bait protein. These additional proteins were still present after 8 washing steps, indicating a strong interaction with NirP1 (Fig. 5b).

Next, we subjected whole-cell lysates and eluate fractions to liquid chromatography-tandem mass spectrometry (LC–MS/MS)-based proteome analysis for the identification of coimmunoprecipitated proteins. In addition to NirP1, a single other protein was strongly enriched, which was identified as the 56 kDa ferredoxin-nitrite reductase (NiR), encoded by *nirA* (*slr0898*) (Fig. 5c). Overall, eight and 72 nonredundant peptides of the NirP1-3xFLAG protein and of NiR were detected, yielding high respective sequence coverages of 94% and 98% (Supplementary Datasets 7, 8, and 9). The combined molecular masses of NirP1 and NiR are compatible with the uppermost signal in the Western blot in Fig. 5b. These results indicated that NiR was a

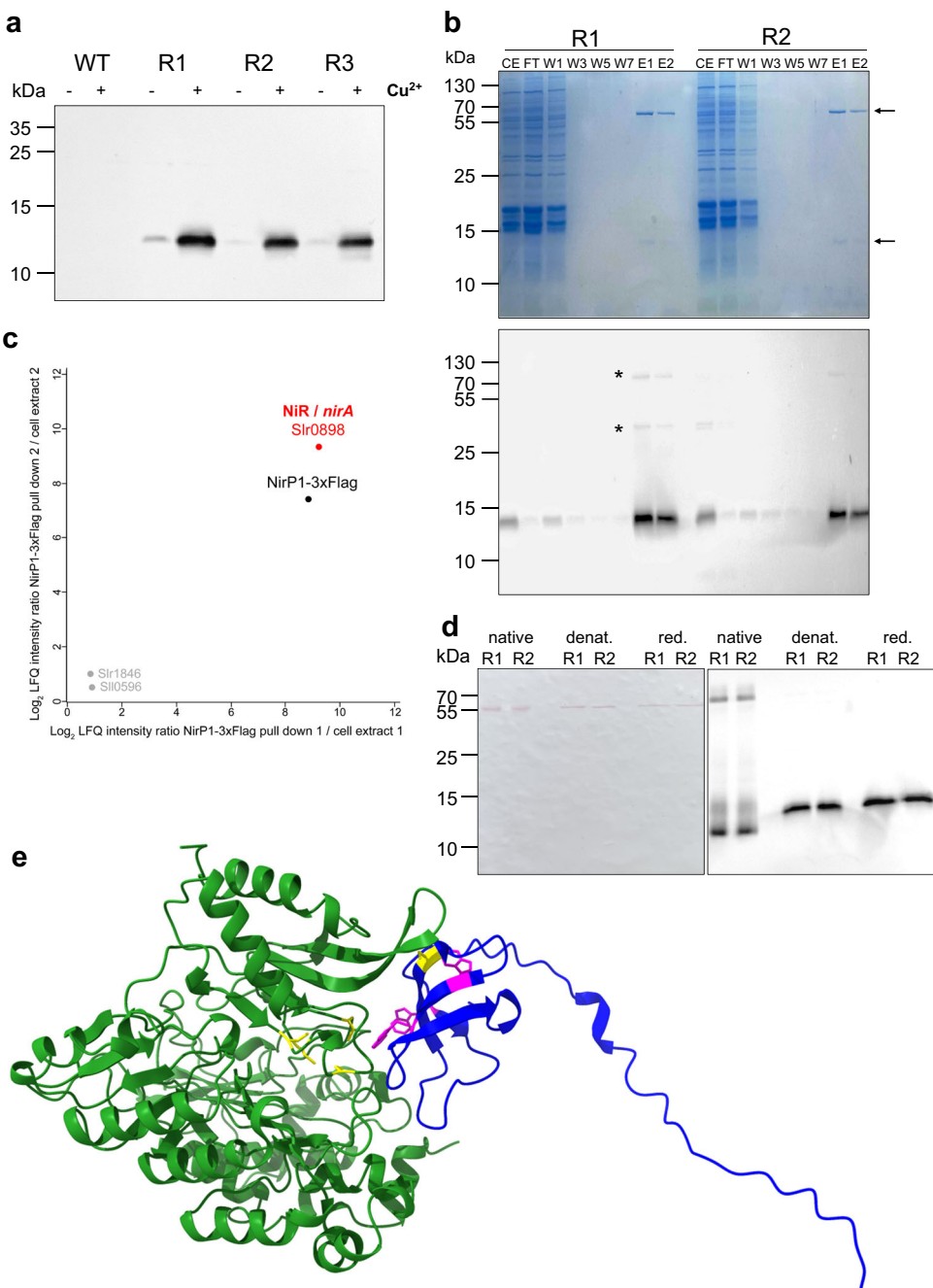

**Fig. 5 | NirP1 expression and pull-down analysis. a** Detection of NirP1 after 24 h of induction with 2 μM Cu₂SO₄ by Western blotting using anti-FLAG antiserum against tagged NirP1 in three biological replicates (R1–R3). The wild type was used as a control. **b** NirP1 pull-down analysis of two biological, independent replicates (R1 and R2). Upper panel: Coomassie-stained SDS gel, the NirP1 signal, and a band indicating a prominent coeluting protein of ~60 kDa are marked with arrows. Fractions are numbered and labeled CE cell extract, FT flow-through, W wash, E elution. Lower panel: Immunological identification of the lower band as NirP1. **c** Mass spectrometry-based analysis of NirP1 co-IP. Scatter plot of log₂-transformed LFQ protein ratios (NirP1-3xFlag co-IP/cell extract) from two independent replicates. Displayed are proteins with a higher abundance in NirP1-3xFlag co-IP elution fractions compared to the initial cell extract. The main interacting protein was identified as ferredoxin-nitrite reductase (NiR), which is encoded by the *nirA/ slr0898* gene. **d** Western blot of NirP1 co-IP fractions using native, denaturing, or reducing conditions. All samples were loaded in biological replicates R1 and R2. **e** NirP1-NiR (WP_010873675.1) interaction in *Synechocystis* 6803 predicted by AlphaFold[31,32] using their full-length sequences. The interaction modeled for the NiR and NirP1 homologs from *Synechococcus* 7942 is shown in Supplementary Fig. S7b. Source data are provided as a Source Data file.

specifically enriched protein that coimmunoprecipitated with the NirP1-3xFLAG protein under the conditions used.

To validate the potential complex formation between NirP1 and NiR, Western blot experiments were performed under different reducing conditions. Under native conditions, two bands appeared. The signal at the lower molecular mass matched the predicted molecular mass of monomeric NirP1, while the upper signal was consistent with the sum of NirP1 and NiR of ~70 kDa. After the samples were denatured by heating or reduced through an addition of dithiothreitol (DTT) and β-mercaptoethanol, the ~70 kDa signal disappeared. The single remaining band migrated higher than the band of the native form, indicating that it was the monomeric and completely unfolded NirP1 protein (Fig. 5d).

Modeling by AlphaFold[31,32] predicted NirP1 binding to the NiR substrate binding pocket next to its catalytic center with the

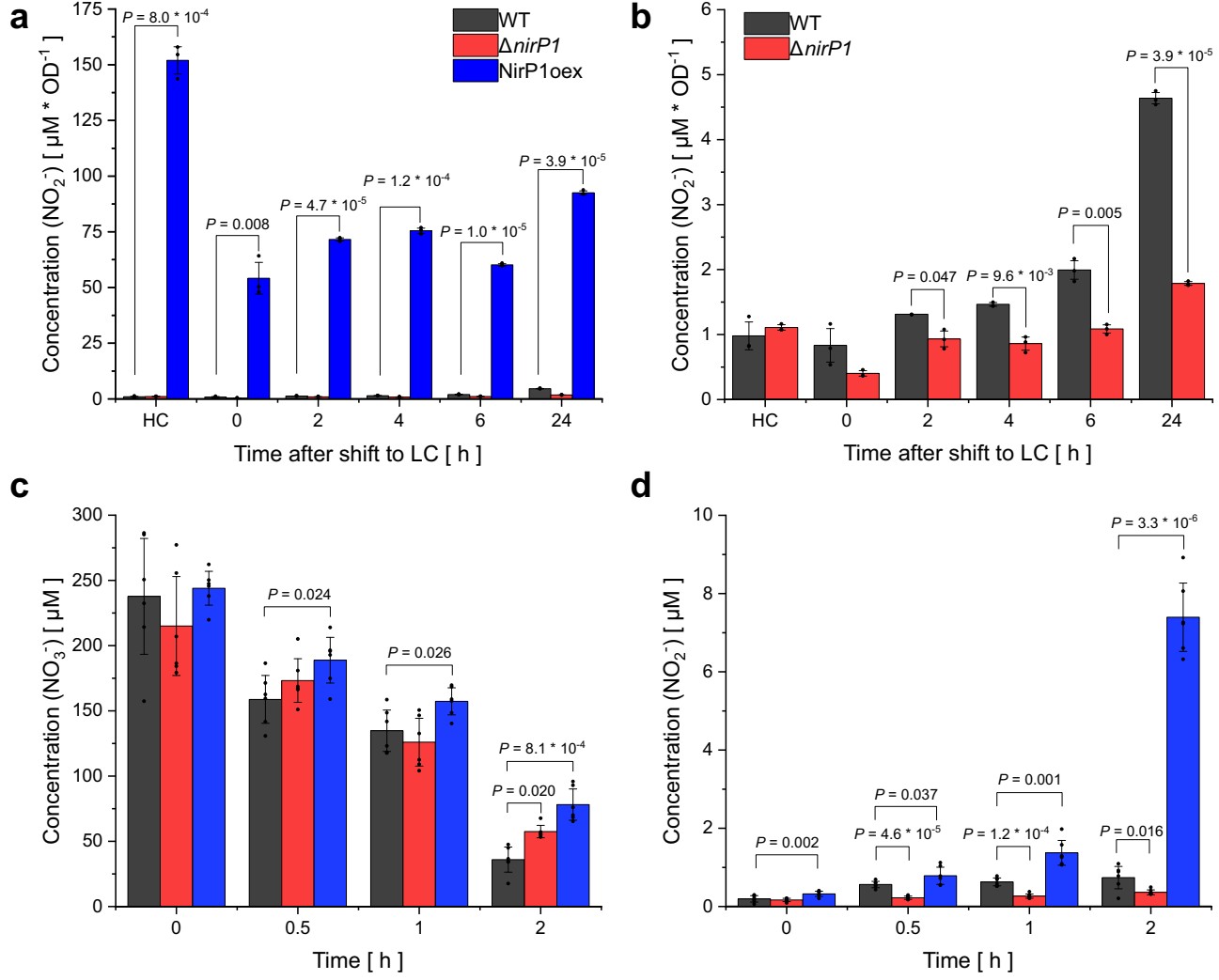

**Fig. 6 | Excretion of nitrite is triggered in the presence of NirP1. a** *Synechocystis* 6803 strains were grown photoautotrophically in BG11 medium with 5% (v/v) $CO_2$ in air as high $C_i$ (HC), harvested and resuspended in new medium and aerated with 0.04% (v/v) $CO_2$ in air to initiate LC conditions. The wild type (WT) started to excrete nitrite in LC conditions when *nirP1* was expressed (see Fig. 1c). **b** Same data as in (**a**) but at different scales to include the results from the NirP1oex strain. Accumulation of nitrite in the supernatant was followed over time. **c** The consumption of nitrate, measured as the decrease in the concentration of nitrate in the supernatants after pelleting the cells by centrifugation. Strains were grown in BG11, washed with N-free BG11 (BG11 −N), set to OD = 1 and resuspended in BG11 containing 200 μM nitrate. **d** The production and excretion of nitrite into the supernatant measured in the same samples as in (**c**) for nitrate. Data are presented as the means ± SD of measuring biological triplicates in duplicates each (*n* = 6). Significance was calculated using a two-tailed *t*-test with unequal variance (Welch's *t*-test; ns no significance; *only *P* < 0.05 are shown) between the strains at corresponding time points. Full details are provided in Supplementary Dataset 10.

iron-sulfur cluster where the reduction of nitrite takes place (Fig. 5e). Interestingly, none of the NirP1 cysteine residues, but two of the three totally conserved tryptophan residues (W36 and W65), were predicted close to the catalytic center. Thus, these amino acids potentially play a functional role.

**Excretion of nitrite is triggered by NirP1**

To test if the action of NirP1 might lead to the excretion of nitrite in LC conditions, as previously reported for *Synechococcus* 7942[9], we first verified if *Synechocystis* 6803 showed this effect as well. Indeed, after 2 h at LC, the *Synechocystis* 6803 wild type, but not the Δ*nirP1* deletion mutant, started to excrete nitrite into the medium (Fig. 6a, b). After 24 h at LC, the concentration of nitrite in the medium of the wild-type strain was ~5 μM. However, in the *nirP1*-overexpressing strain the nitrite concentration was much higher with ~75 μM (Fig. 6a, b). All three results (Δ*nirP1*, wild type, NirP1oex) suggested that the excretion of nitrite into the supernatant was related to the presence of NirP1 in the cell. This effect was even more pronounced by the ectopic

overexpression of NirP1 in BG11 medium aerated with 5% (v/v) $CO_2$ leading to an ~150-fold higher accumulation of nitrite in the supernatant of NirP1oex compared to wild type and Δ*nirP1* grown samples (HC in Fig. 6a). After the cells were washed and resuspended in fresh medium, the overexpressor immediately excreted nitrite again (time point 0 in Fig. 6a). These data establish that NirP1 triggers the excretion of nitrite.

To ensure that the accumulated nitrite was a product of absorbed and reduced nitrate, the uptake of nitrate and subsequent excretion of nitrite were measured in parallel. The cultures were first grown in N-free BG11 medium and then 200 μM nitrate was added. In the control experiment, an additional 2 mM ammonia was added because nitrate uptake in *Synechocystis* is inhibited in the presence of ammonia and the transcription of nitrate and nitrite reductase genes is repressed. After 2 h, the overexpressor led to significant differences in nitrite concentration in the medium compared to wild type and Δ*nirP1*. In addition, the overexpression seemed to slow the uptake of nitrate (Fig. 6c, d). Here, the wild type, although less pronounced than the

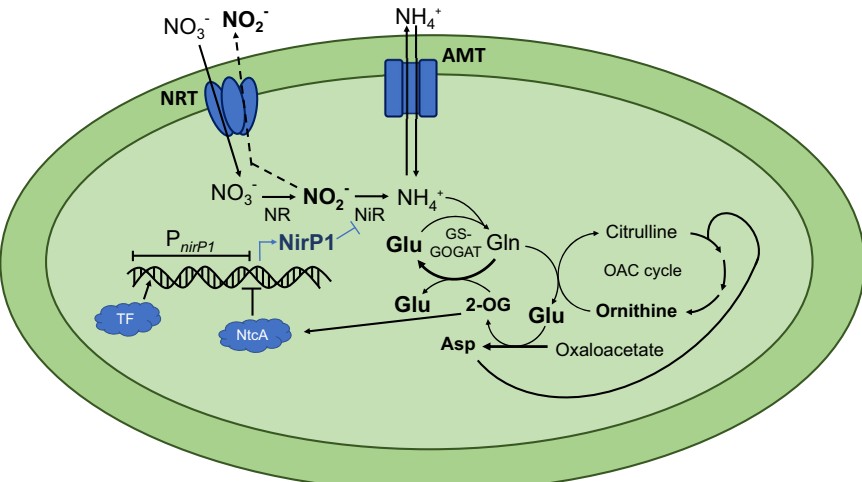

**Fig. 7 | NirP1 is a regulator at a central position in the assimilation of inorganic nitrogen.** Cyanobacteria can assimilate N from simple compounds, such as ammonium, nitrate, nitrite, and urea. During nitrate assimilation, intracellular nitrate is reduced in two steps to nitrite and ammonium, which is then incorporated into glutamate in the GS/GOGAT cycle or into carbon skeletons during the synthesis of amino acids by transaminases. The reduction of nitrate to nitrite is catalyzed by the NR and the subsequent reduction to ammonium by the NiR. Transcription of the regulatory factor NirP1 is controlled by an unknown transcription factor (TF) and by NtcA. 2-OG is a corepressor of NtcA. Therefore, an increased 2-OG level leads to the repression of *nirP1* transcription, while a low level of 2-OG leads to its activation. NirP1 binds to the enzyme nitrite reductase (NiR), leading to the inhibition of nitrite reduction. The accumulating nitrite is then rapidly exported by an unidentified export system. Transport systems for ammonium (AMT) and nitrate/nitrite (NRT) uptake are shown.

overexpressor, also excreted nitrite into the medium, as observed in the LC shift experiment before. In contrast, in the control experiment with additional ammonium, no nitrate uptake and no excretion of nitrite into the medium were observed (Supplementary Fig. S6a, b). These results demonstrated that the accumulation of nitrite in the medium of the wild type and overexpression strains resulted from nitrate uptake, the subsequent reduction to nitrite by NR and the alteration of NiR activity, with the involvement by NirP1.

## Discussion

Nitrate is one of the most abundant N-containing nutrients for cyanobacteria, while the preferred source for nitrogen assimilation is ammonium. Therefore, complex regulatory mechanisms occur to regulate nitrogen assimilation and uptake, e.g., genes of nitrate assimilation are repressed if ammonium is available[37,38]. When nitrate is the only N source available, it is taken up and reduced in two steps to nitrite and ammonium, which is then incorporated into glutamate in the GS/GOGAT cycle (Fig. 7). The reduction of nitrate to nitrite is catalyzed by the enzyme NR and the subsequent reduction to ammonium by NiR. NirP1, a small regulatory protein, is upregulated in LC conditions in *Synechocystis* 6803 (Fig. 1a). Its elevated expression, as shown in this study, leads to the accumulation of nitrite in the medium. We identified NiR as the primary binding partner for NirP1 (Fig. 5c). Therefore, the results can be explained if the NirP1:NiR interaction directly leads to the accumulation of nitrite in the cell, which is toxic and therefore becomes rapidly exported (Fig. 7). Excess internal nitrite has been shown to inhibit photosynthesis, particularly the activity of photosystem II in *Synechocystis* 6803[39]. Comparable responses were reported for N-starved *Synechocystis* 6803 mutants defective in glycogen synthesis that excreted 2-OG and pyruvate[34].

Several factors control primary N metabolism, such as the transcription factor NtcA or the $P_{II}$ regulator and proteins interacting with them[10–14,17–19]. However, unlike several other regulatory proteins that modulate the activity of other regulators (e.g., PirA and PipX), NirP1 targets one of the key enzymes of N assimilation metabolism directly. This is reminiscent of the direct inhibitory role of the small GS inhibitory factors IF7 and IF17[17]. These findings suggest that a higher number of relatively small proteins might exist that

modulate or modify the activity of relevant enzymes and await characterization.

To examine the possible mode of action, we further predicted the interactions between NiR and other proteins. In *Synechocystis* 6803, ferredoxin Fd1 (*ssl0020*, *petF*) acts as an electron donor for the reduction of nitrite[40]. Therefore, we modeled the interaction of NiR with NirP1 (Fig. 5e) and Fd1 (Supplementary Fig. S7a). The results suggested that both proteins compete for the binding pocket of NiR. By binding NirP1 instead of Fd1, the binding pocket is blocked for electron transfer and the subsequent reduction of nitrite to ammonium. Thus, this could be a possible mechanism by which NirP1 inhibits NiR, leading to the excretion and accumulation of nitrite in the medium during NirP1 expression.

Transcription of *nirP1* is controlled by an unknown transcription factor and by NtcA. If 2-OG cannot be aminated efficiently in the GS-GOGAT cycle due to N starvation, $C_i$ oversupply, or a combination of both, it will accumulate in the cell. The increased 2-OG level is sensed by PII and transferred to NtcA, leading to stronger promoter binding and the repression of *nirP1* transcription. Therefore, NirP1 is at the confluence of C and N assimilation and is effectively a regulator of the C/N balance.

In addition to the NtcA-binding site, we found another putative promoter element at position −43 to −60 (Fig. 2a), the mutation of which rendered the *nirP1* promoter inactive. Therefore, this element is likely recognized by a positively acting transcription factor, although its identity was not further investigated in this work and should be studied in the future.

It was previously reported that during photoautotrophic growth in shifts to LC conditions, *Synechococcus* 7942 excreted nitrite into the medium[9]. As the fundamental reason for this observation, a decline in the enzymatic activity of NiR was identified that started within 30 min. This decline was not a consequence of changed *nirA* expression because the observed decrease in NiR activity under LC conditions was substantially larger than that observed when the cultures were treated with rifampicin to inhibit transcription. Instead, a mechanism was postulated that directly inactivated NiR under LC conditions[9]. Our data were obtained in *Synechocystis* 6803, but a close NirP1 homolog is found in *Synechococcus* 7942 (Fig. 1b), and an almost identical

interaction with NiR was predicted (Supplementary Fig. S7b, c). Therefore, we have likely identified the mechanism underlying previously observed NiR inhibition and nitrite excretion. We calculated that approximately 5% of assimilated nitrate was excreted as nitrite into the medium by *Synechocystis* 6803 in the first 2 h (Fig. 6c, d). Compared to *Prochlorococcus* strains MIT0917 and SB, and *Synechococcus* 7942, which were reported to excrete up to 30% of the assimilated nitrate as nitrite[6,9], the here-measured nitrite excretion of *Synechocystis* 6803 was less pronounced. Previous studies reported for *Synechocystis* 6803 a higher excretion, of up to 60% of the reduced nitrate under mixotrophic conditions[41], which clearly shows that *Synechocystis* 6803 is capable to release high nitrite amounts into the medium. The *nirP1* overexpression strain was obviously not significantly inhibited by the remaining nitrite, which could indicate that it had a way to avoid nitrite poisoning by storing it at a physiologically neutral place (or the regulation by NirP1 was not completely inhibiting the NiR activity). Indeed, there is circumstantial evidence for such a nitrite storage, possibly in the periplasm. We observed a very fast immediate release of nitrite from washed cells of the NirP1ex strain (see time point 0 in Fig. 6a), which then was followed only by a slower further increase. Thus, under photoautotrophic conditions used here, the cells may have a mechanism to store most of the nitrite in order to resume normal N assimilation when the $C_i$ supply increases again. This points at a possible mechanism for deciding whether nitrite is released or kept within the cell, depending on the physiological conditions. These points are interesting topics for future research.

According to our data, NirP1 expression is upregulated under LC conditions and then participated in the inhibition of NiR, leading to nitrite excretion and accumulation. Surprisingly, the regulation occurs at an unexpected point (Fig. 7), as controlling N assimilation during nitrate uptake or during nitrate reduction seems more intuitive. Here, uncontrolled nitrate uptake led to light-dependent nitrite excretion under low light conditions[42]. Therefore, the availability of electrons could play a role because the reduction of nitrite requires six moles of reduced ferredoxin, while the reduction of nitrate requires only two. Another consideration is the possible beneficial effect of nitrite excretion on cooccurring microbes in the natural environment. Certain *Prochlorococcus* release up to 30% of their N uptake as extracellular nitrite when grown on nitrate[6]. *Prochlorococcus* typically contains the most compact genomes of any free-living cyanobacteria, which was linked to its quasi-homeostatic environment[43–45]. Because we did not identify NirP1 homologs in *Prochlorococcus*, the underlying mechanism probably operates without NirP1. In other strains of cyanobacteria, exposed to greater variations in the environment, the involvement of NirP1 has likely led to a more accurate regulation. In any case, the release of an N cycle intermediate, such as nitrite, should have tremendous relevance for certain other members of the microbiome. Our results show that homologs of NirP1 are widely distributed among cyanobacteria, expanding our understanding of how C and N metabolism become integrated in these organisms.

## Methods

### Cultivation conditions and spectrometric measurements

Strains were maintained in BG11 or in copper-free BG11[28] supplemented with 20 mM TES pH 7.5 under continuous white light of 50 μmol photons $m^{-2} s^{-1}$ at 30 °C. Mutant strains containing pVZ322-derived plasmids were cultivated in the presence of 50 μg/mL kanamycin, while Δ*nirP1* was maintained at 20 μg/mL streptomycin. The cultures for overexpression of NirP1 using the $P_{petE}$ promoter for pull-down experiments and MS measurements were grown either in BG11 or under CellDeg® conditions supplemented with 2 μM CuSO₄. Cultivation with CellDeg® conditions was performed for 4 days in 100 mL FOM (freshwater organism medium; 50 mM NaNO₃, 15 mM KNO₃, 2 mM MgSO₄, 0.5 mM CaCl₂, 0.025 mM H₃BO₃, supplemented with 0.15 mM FeCl₃ /Na₂EDTA, 1.6 mM KH₂PO₄, 2.4 mM K₂HPO₄, 10 mM

NaHCO₃, 10 nM MnCl₂, 0.1 nM ZnSO₄, 2 nM Na₂MoO₄, 10 pM CuSO₄, and 30 pM CoCl₂) inoculated with culture with a starting OD of ~0.8 and 150 μmol photons $m^{-2} s^{-1}$. High CO₂ concentrations were delivered to the flasks from a carbonate buffer (0.3 M K₂CO₃ and 2.7 M KHCO₃) through a highly gas-permeable polypropylene membrane. Light intensity was increased during growth according to the cell density. 24 h prior harvesting the cells (5000 × $g$, 4 °C, 10 min), protein expression was started by adding 2 μM CuSO₄. Cultures in growth experiments and phenotypic assays were grown in BG11 medium supplemented with copper but without antibiotics to prevent any possible side effects.

Whole-cell absorption spectra were measured using a Specord® 250 Plus (Analytik Jena) spectrophotometer at room temperature and were normalized to the wild type at 750 nm. Cultures were measured in triplicate. Cultures at an $OD_{750} > 1$ were diluted with 1× BG11 prior to taking the absorption spectra.

### Construction of mutant and overexpression lines

*Synechocystis* 6803 PCC-M[46] was used as the wild type and background for the construction of mutants. Knockout mutants were generated by replacing the *nirP1* (*ncr1070*) coding sequence with a streptomycin resistance cassette (*aadA*)[47] by homologous recombination and using pUC19 as a vector for subcloning. The construct for gene replacement was generated by AQUA cloning[48]. Details of primer sequences and plasmids are provided in Supplementary Tables S1 and S2. Segregation of *nirP1* and Δ*nirP1/aadA* alleles was checked by colony PCR using the primers P_AK9 and P_AK10 (Supplementary Fig. S1).

To complement the knockout and create a NirP1-overexpressing strain, *nirP1* was inserted into the self-replicating pVZ322 plasmid under control of the Cu²⁺-inducible $P_{petE}$ promoter or under control of its native promoter ($P_{nirP1}$). The inserts for the overexpression of *nirP1* were assembled into pVZ322 predigested by *Xba*I and *Pst*I for 16 h at 37 °C by AQUA cloning[48]. The constructs were checked using the primers P_AK16 and P_AK17 (Supplementary Fig. S1). The sequence-verified plasmids were introduced into *Synechocystis* 6803 by electroporation[49]. For the electroporation, culture was cultivated to exponential phase with OD ~0.7–1.0, 25 mL washed three times with ice-cold HEPES buffer (1 mM, pH = 7.5) and resuspended in 0.5 mL HEPES buffer. 1 μg of plasmid was mixed with 100 μL of cells and transferred into a precooled Gene Pulser/MicroPulser Electroporation Cuvettes (0.2 cm gap, BIO-RAD). A pulse of 12.5 kV and 4 ms was executed (MicroPulser Electroporator, BIO-RAD), 1 mL of BG11 (BG11 lacking copper for constructs using the *petE* promotor) was added to the cells and transferred to 50 mL BG11 for 24 h cultivation under continuous light of 50 μmol photons $m^{-2} s^{-1}$ at 30 °C. The culture was harvested (5000 × $g$, 4 °C, 10 min), resuspended in 100 μL BG11, and plated on 0.75% Kobe-agar plate with 50 μg/mL kanamycin and cultivated for 5 days. Single colonies were transferred on a new plate and verified by PCR (Supplementary Fig. S1, Supplementary Table S1). The expression of *nirP1* was checked by Western hybridization using ANTI-FLAG antisera (Sigma–Aldrich, #SAB4301135).

### RNA preparation and northern blot verification

Cultures were grown in BG11 supplemented with 10 mM NHCO₃ (HC), washed three times with BG11 lacking any $C_i$ source (LC), and grown for 1, 3, 6, 12, and 24 h in LC conditions to induce *nirP1* expression. The time immediately before $C_i$ removal (0 h, HC) served as the negative control. Cells were collected by filtration through hydrophilic polyethersulfone filters (Pall Supor®–800, 0.8 μm), transferred into PGTX buffer, and instantly frozen in liquid N₂[50]. Samples were incubated for 10 min at 65 °C in a water bath, one volume chloroform:isoamyl alcohol (24:1) was added, and were incubated for 15 min at room temperature with several vortexing cycles. After centrifugation for 3 min at 3250 × $g$, the supernatant was transferred to a fresh tube, one volume of chloroform:isoamyl alcohol was added. This step was repeated

twice. RNA was precipitated by the addition of one volume of iso-propanol and incubated for 16 h at −20 °C. For northern blot verification, 10 µg of RNA per well was loaded on a 10% polyacrylamide-8.3 M urea gel and electroblotted onto Hybond N nylon+ membranes (Amersham) with 1 mA per cm$^2$ for 1 h. *nirP1* transcript accumulation was analyzed by Northern hybridization using single-stranded radio-actively labeled RNA probes transcribed in vitro from PCR-generated templates using primer P_AK30 and P_AK31 (Supplementary Table S1). The radioactively labeled probes were generated using [α-$^{32}$P]-UTP and the Maxiscript T7 In vitro transcription kit (Thermo Fisher Scientific). Hybridizations were performed in 50% deionized formamide, 7% SDS, 250 mM NaCl, and 120 mM Na$_2$HPO$_4$/NaH$_2$PO$_4$ pH 7.2 overnight at 62 °C. The membranes were washed in buffer 1 (2× SSC (3 M NaCl, 0.3 M sodium citrate, pH 7.0), 1% SDS), buffer 2 (1× SSC, 0.5% SDS) and buffer 3 (0.1× SSC, 0.1% SDS) each for 10 min. All wash steps were performed at 57 °C. Signals were detected with a Laser Scanner Typhoon FLA 9500 (GE Healthcare). Signal intensities of *nirP1* were determined using Quantity One software (BIO-RAD) and normalized to the signal of the 5S rRNA (Fig. 1c).

## Promoter analysis

To verify *nirP1* promoter activity in vivo in *Synechocystis* 6803, the nucleotides −70 to +30 (TSS at +1) were fused to *luxAB* reporter genes and integrated into the genome. Initially, cultures were set to an OD 0.7 and grown under standard BG11 or HC conditions for 2 h and subsequently transferred to NO$_3^-$- or NaCO$_3$-free BG11 or BG11 medium supplemented with additional NH$_4$Cl to induce transcription. Measurements were carried out for 24 h in LC or high NH$_4$Cl conditions or for 7 days under N deprivation. To recover from N starvation, the bleached cells were transferred back to standard BG11 medium after 7 days and observed for 3 days. Bioluminescence was measured in vivo as total light counts per second by using a VICTOR$^3$ multiplate reader (PerkinElmer). Prior to the measurement, cells were diluted to an OD$_{750}$ = 0.7 and 200 µL of the suspension were filled into a white 96-well plate (CulturePlate™-96, PerkinElmer). To increase the signal, 2 µL of Decanal was added to the samples. In the multiplate reader the cell suspensions were shaken for 10 s and subsequently total light emission was measured for 1 s. A strain carrying the promotorless *luxAB* genes served as a negative control[51]. The raw data, the calculation, and statistical evaluations are provided in Supplementary Dataset 3.

## Co-IP, LC-MS/MS analyses, and data processing

The overexpression of NirP1-3xFLAG was induced in exponentially growing cultures of the *Synechocystis* 6803 strain NirP1oex in the Cell-Deg system by adding Cu$_2$SO$_4$ to a final concentration of 2 µM. 50 mL of cells were harvested by centrifugation (5000 × *g*, 4 °C, 10 min) after 24 h of NirP1-3xFLAG overexpression, resuspended in 2 mL FLAG buffer (50 mM HEPES-NaOH pH 7; 5 mM MgCl$_2$; 25 mM CaCl$_2$, 150 mM NaCl; 10% glycerol; 0.1% Tween-20) supplemented with Protease Inhibitor (cOmplete, Roche) mixed with 100 µL of glass beads (Ø 0.1–0.25 mm, Retsch) and lysed mechanically in a prechilled Precelly24 (Bertin Instruments) with three rounds of shaking at 6000 rpm (10 s) followed by a pause (5 s) per cycle, for a total of 5 cycles. To remove cell debris and glass beads, extracts were centrifuged (1000 × *g*, 5 min, 4 °C). Supernatants were collected for further processing. Remaining cell debris was removed by centrifugation (15,000 × *g*, 4 °C, 30 min). The cleared total cell lysate was subjected to co-IP using Anti-FLAG M2 affinity magnetic beads (Sigma−Aldrich) rotating for 1 h at 4 °C. The beads were washed 8 times with FLAG buffer (2 min, 4 °C, rotating). Bound proteins were eluted with FLAG peptide (Sigma−Aldrich, 100 ng/µL) in TBS buffer (150 mM NaCl in 20 mM Tris/HCl, pH = 7.6) at 4 °C. Two independent replicates were prepared. Protein concentrations of eluate fractions and respective whole-cell lysates were measured in the Coomassie Plus Bradford Assay (Thermo Fisher Scientific) through the plate photo reader 1420 multilabel counter (PerkinElmer) in duplicate, and 500 µL of

each was subsequently snap frozen. Proteins in whole-cell lysates were purified by acetone/methanol precipitation[52], and co-IP eluates were lyophilized. For further analysis, samples were resolubilized in denaturation buffer (6 M urea, 2 M thiourea in 100 mM Tris/HCl; pH 8.0) at a final protein concentration of 1 µg/µL. Protein disulfide bonds were reduced with 1 mM dithiothreitol for 45 min and the resulting thiol groups were alkylated with iodoacetamide at a final concentration of 5.5 mM for 45 min. Predigestion of proteins with Lys-C (Santa Cruz Biotechnology) for 3 h was followed by dilution with 4 volumes of 20 mM ammonium bicarbonate buffer, pH 8, and overnight digestion with trypsin (sequencing grade, Promega), both at protease/protein ratios of 1/100. The resulting peptide solutions were acidified with tri-fluoroacetic acid to pH 2.5, and an aliquot corresponding to 10 µg peptides was purified by stage tips[53]. For LC–MS/MS-based protein analysis, 40 ng of each sample was loaded onto an in-house-made 20 cm column with 75 µm ID, packed with ReproSil-Pur 1.9 µm C18 material (Dr. Maisch, Germany) and separated by RP chromatography on an EASY-nLC 1200 system (Thermo Fisher Scientific, USA) using 60 min gradients. Eluting peptides were analyzed on an Orbitrap Exploris 480 mass spectrometer (Thermo Fisher Scientific, USA) as described elsewhere[54].

All raw spectra were processed with MaxQuant software (version 1.6.8.0)[55] at default settings. The match between run option and label-free quantification (LFQ) was enabled. Peak lists were searched against an in-house modified target-decoy database of *Synechocystis* 6803 with 3,681 protein sequences, including the NirP1-3xFLAG fusion protein sequence. False discovery rates were limited to 1% at the peptide and protein levels. LFQ protein intensities of two independent co-IPs were normalized by corresponding values from whole-cell extract measurements, log$_2$ transformed, and plotted against each other using the Perseus software suite (version 1.6.5.0)[56].

## SDS–PAGE, native gel electrophoresis and western blotting

Proteins were mixed with denaturing or reducing loading dye (5× concentrated: 250 mM Tris-HCl pH 6.8, 25% glycerol, 10% SDS, 500 mM DTT, and 0.05% bromophenol blue G-250). Moreover, 6% ß-mercaptoethanol (v/v) was added fresh to the sample in reducing conditions before heating for 5 min at 95 °C. The protein samples were separated by 12% SDS–PAGE using the PageRuler™ molecular weight marker (Thermo Fisher). To run samples under nonreducing conditions, a native loading dye (5× concentrated: 30% glycerol, 0.05% bromophenol blue G-250, 150 mM Tris-HCl pH 7.5) was used, and boiling was omitted. To make proteins visible, the gel was stained with Coomassie dye. The gel was incubated for 30 min under slight shaking, covered with Instant Blue Coomassie Protein Stain (Abcam), and washed overnight with ddH$_2$O. For immunoblot analysis, separated proteins were transferred to nitro-cellulose blotting membrane (Amersham™ Protan, pore size 0.4 5 µm). Membranes were blocked overnight at 4 °C with 5% milk powder in TBS-T, washed three times (10 min), and subsequently probed with monoclonal ANTI-FLAG® M2-Peroxidase (HRP) antibody (Sigma−Aldrich, #SAB4301135 at a dilution of 1:5000) in TBS-T for 1 h at room temperature in the dark and washed again three times (10 min). All washing steps were performed with TBS-T (20 mM Tris pH 7.6, 150 mM NaCl, 0.1% (v/v) Tween-20) at room temperature. Signals were detected with WesternBright™ ECL-spray (Advansta) on a chemiluminescence imager system (Fusion SL, Vilber Lourmat) and subsequently visualized using FUSION-CAP (Vilber Lourmat) and Quantity One software (BIO-RAD).

## Nitrate and nitrite measurements

Wild type, Δ*nirP1*, and *nirP1*-overexpressing cultures were cultivated photoautotrophically in BG11 medium with 5% (v/v) CO$_2$ in air as the high CO$_2$ condition (HC), harvested and resuspended in new medium and aerated with 0.04% (v/v) CO$_2$ in air to start low CO$_2$ (LC) conditions. Samples were taken under HC conditions before cells were shifted to LC conditions and then after 0, 2, 4, 6, and 24 h of growth in LC conditions and measured in biological triplicates.

To measure nitrate uptake and nitrite accumulation, cells were washed three times and resuspended in BG11 without NaNO$_3$ (BG11-N) lacking CuSO$_4$. After incubation for 1 h, only 200 μM NaNO$_3$ as the nitrogen source or NaNO$_3$ together with 2 mM NH$_4$Cl and a final concentration of 2 μM CuSO$_4$ were added. Samples were collected at the indicated time points and centrifuged (16,800 × $g$, 5 min, room temperature). Concentrations in the resulting supernatants were determined by ion chromatography measurements or with the Griess Reagent Kit (Promega). For IC measurements, samples were mixed with methanol (1:3 (v:v)) and centrifuged three times (16,800 × $g$, 10 min, room temperature), and 10 μL was loaded and separated by a column (DionexTM IonPacTM AS11-HC, Thermo Fischer). Ions were identified, and concentrations were quantified using the area (μS min$^{-1}$) given in the software (Chromeleon® 7). Standards at various concentrations were used for calibration. The raw data, calculations, and statistical evaluations are given in Supplementary Dataset 10. Nitrite was measured after conversion into an azo dye[57]. The determination limit was 0.09 μmol L$^{-1}$. The combined standard uncertainty was 4.9%. Samples higher than the measuring range were diluted with ultrapure water.

### Extraction and metabolite analysis

Wild type, Δ$nirP1$, and $nirP1$ overexpressor cultures were set in duplicates or triplicates to an OD of 0.5 and grown for 24 h without antibiotics before cells were pelleted by centrifugation (3,210 × $g$, 10 min, room temperature), washed three times in copper-free BG11 medium and induction of NirP1 was started by adding 2 μM CuSO$_4$ after 1 h. After 0, 1, 3, 12, and 24 h of incubation, 2 mL of each culture was collected by centrifugation (16,800 × $g$, 5 min, room temperature) and immersed in liquid N$_2$. For extraction, 630 μL of methanol containing carnitine (internal standard, 1 mg per extraction) was added, mixed for 1 min, and incubated for 5 min in a sonic water bath. After another 15 min of shaking at room temperature, 400 μL of chloroform was added, and the sample was incubated at 37 °C for 5 min. Next, 800 μL of ROTISOLV LC–MS grade H$_2$O (Roth) was added, and the sample was mixed thoroughly. Precipitation was enabled by storage at 22 °C for at least 2 h, and phase separation was subsequently achieved by centrifugation for 5 min at room temperature (16,800 × $g$). The upper polar phase was collected and dried in a speed vac for 30 min followed by lyophilization overnight.

Absolute metabolite contents were quantified on a high-performance liquid chromatography–mass spectrometer LC–MS-8050 system (Shimadzu)[58]. Dried samples were dissolved in 200 μL LC–MS grade water and filtered through 0.2-mm filters (Omnifix-F, Braun, Germany), and 5 μL of the cleared supernatant was separated on a pentafluorophenylpropyl column (Supelco Discovery HS FS, 3 mm, 150 3 2.1 mm).

The compounds were identified and quantified using the multiple reaction monitoring (MRM) values given in the LC–MS/MS method package and the LabSolutions software package (Shimadzu). Authentic standard substances (Merck) at various concentrations were used for calibration, and peak areas were normalized to signals of the internal standard (carnitine). The data were further normalized to the OD$_{750}$ and volume measured for each sample. The raw data and estimations as absolute values in ng per mL per OD$_{750}$ are given in Supplementary Datasets 5 and 6, with all the statistical evaluations.

### Reporting summary

Further information on research design is available in the Nature Portfolio Reporting Summary linked to this article.

## Data availability

The mass spectrometry proteomics data generated in this study have been deposited at the ProteomeXchange Consortium via the PRIDE[59] partner repository with the dataset identifier PXD041127. Source data are provided with this paper.

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

## Acknowledgements

We are very grateful to Vanessa Krauspe for the previous work on the identification of novel small proteins, Marcus Ziemann for the support in bioinformatic analysis, and Annegret Wilde (all Freiburg) for helpful discussions and for access to the spectrophotometer. We thank Ralf Weßbecher (AG Boll, Freiburg) for access and help during the ion chromatography measurements. We appreciate the support by the Deutsche Forschungsgemeinschaft (DFG, German Research Foundation) through the FOR2816 research group SCyCode to A.K., P.S., M.H., B.M. and W.R.H. (grants HA 2002/23-2, MA 4918/4-2, and HE 2544/15-2).

The LC–MS/MS equipment at the University of Rostock was financed through the Hochschulbauförderungsgesetz program (GZ: INST 264/125-1 FUGG) to M.H.

## Author contributions

WRH designed the project and secured funding. Construction of mutant and overexpressing strains, characterization of the strains, and co-IP experiments were performed by AW and AK. PS and BM performed and interpreted the MS-based proteomic analyses. Metabolite levels were analyzed by ST and MH. RS performed the shift to LC and the corresponding measurements of nitrite levels in the medium. All other experiments and analyses were performed by AK. AK and WRH wrote the manuscript with input from all authors.

## Funding

## Competing interests

The authors declare no competing interests.
