## [Peer Review File · Nature Communications]

Protein NirP1 regulates nitrite reductase and nitrite excretion
in cyanobacteriaReviewer #1 (Remarks to the Author):

The key results described by Kraus et al., provide evidence of a new mechanism by which nitrite is released from cyanobacteria during growth on nitrate. The authors present an elegant series of experiments that implicate the NirP1 protein in modulating nitrite release by *Synechocystis* by interacting with the nitrite reductase protein NirA. Further, they demonstrate that this NirP1 protein is, itself, likely regulated by NtcA (via repression) and an additional unknown activator. It appears that NirP1 may allow for more precise control over C/N homeostasis within the cyanobacteria that encode this post-translational regulatory protein in their genomes. The hypotheses, methods, statistical analyses, interpretations, and conclusions appear to be robust. I think the work is highly significant given that it is still unclear to the scientific community why some cyanobacteria release nitrite during growth on nitrate (particularly under nutrient limiting conditions). The authors supply some valuable new information on this front that is related to C-limitation in cyanobacteria and manifested via post-translation regulation of nitrite reductase.

Suggested Improvements:

(1) It would be very useful if Supplemental Dataset 1 was modified to provide information on the organism represented by the sequence record. This would facilitate access to both casual readers and those who want to further explore this phenomenon across cyanobacteria.

(2) Regarding the growth response of the overexpression mutants on nitrate, the data are convincing that there is a growth defect (as well as enhanced nitrite production during growth on nitrate as presented elsewhere in the manuscript). I was, however, curious about whether or not a growth defect (as well as pigmentation change) would be apparent if the nitrate was replaced entirely by ammonium as the N source. I would hypothesize that the the growth and pigmentation defects would be entirely or nearly eliminated given that the nitrate to ammonium reduction pathway would be completely bypassed during growth on ammonium. While this experiment would provide further support for the proposed role of NirP1, the data already provided are quite sufficient. Yet, if the authors already have that data available, I think it would be a nice addition.

(3) L263: I think you mean glutamine synthetase rather than GOGAT.

(4) Fig. 3: I would appreciate the inclusion of growth rates either overlaid on the figure or provided in a table or within the legend. I see that there is certainly a pronounced lag phase, but it's not immediately clear how different the growth rates are during log-linear growth.

(5) Data availability: Thank you for submitting the proteomics data to a repository. Have the authors also considered submitting the data on nitrate consumption, nitrite production, and OD values to a data repository?

(6) Clarity and context: Overall, the writing is clear and easy to follow. But the authors might want to consult with Nat Comm on which acronyms should be defined. I recommend rephrasing the sentence on lines 326-327 for clarity given its confusing syntax: "However, this concentration was with ~75 μ M much higher for the nirP1-overexpressing strain".

Reviewer #2 (Remarks to the Author):

This paper describes a very nice study focused on a previously unknown gene they call nirP1 in *Synechocystis* PCC 6803. The authors used several approaches to investigate the function of this protein: they show that part of a previously observed ncRNA actually encodes a small protein, highly conserved in cyanobacterial genomes (with the exception of *Prochlorococcus* and symbiotic strains as UCYN-A). nirP1 transcription is controlled by NtcA and another transcriptional factor, being expressed under low carbon conditions. Deletion or ectopic expression of this gene lead to clear effects on the growth and color of cultures. The concentration of some nitrogen-related metabolites (in particular, glutamate) was affected in mutants overexpressing nirP1. Co-

immunoprecipitation and Western blot experiments showed NirP1 binds to nitrite reductase, suggesting it is a regulator (inhibitor) of this enzyme. Finally, the authors observed that wild type *Synechocystis* 6803 does excrete nitrite under low carbon conditions, while the nirP1 deletion mutant does not, as a consequence of nitrite reductase inhibition. They also demonstrate that the accumulated nitrite is a result from nitrate uptake.

This manuscript contains a substantial amount of work, and the experimental design has been carefully carried out, addressing some of the main questions on the function of this gene. In my opinion it provides a significant advance in the biology of cyanobacterial metabolism and its regulation. Hereafter I suggest some ideas for improvement:

1. The results shown in the paper demonstrate that NirP1 is a small protein involved in the regulation of nitrite reductase in *Synechocystis* PCC 6803. This results in nitrite excretion (although in uncertain ways, which I will discuss later). Hence I think the title should be based on the concept of nitrite reductase binding/regulation by NirP1, which is the core of the paper, rather than nitrite excretion.
2. Lines 137-144: The authors say that nirP1 is present in genomes from all morphological subsections of cyanobacteria, with the exception of *Prochlorococcus* and UCYN-A. I wonder if they checked the genomes of all available marine strains of *Synechococcus*, and if so, do all of them possess this gene?
3. Line 155: The text says that the strongest signal was obtained 1 h after the shift "with somewhat declining intensity at the later time points". However, the signal corresponding to 24 h is also strong, which does not fit with the description of results. Could the authors provide some explanation for such a high signal after 24 h?
4. Line 162: The text says "single detected signal" of ca. 12 kDa in Fig. 1D, but there are actually two bands in this figure, which in samples R1 and R2 without NHCO_3 are especially remarkable. I missed an explanation for the presence of this second band (located above) and why it is more intense in the samples without NHCO_3 .
5. Lines 184-186: Although Figure 2 shows clearly two mutated versions of the promoter, the text is somehow confusing and the reader might believe there is a single mutated promoter with changes in both the NtcA binding site and in the upstream element; I suggest changing the text for clarity.
6. Lines 195-196: Did the authors perform any experiment to assess the identity of this second transcription factor possibly involved in the control of nirP1? In any case, it might be interesting to speculate on the possibilities in the discussion.
7. The results shown in panel 1C are quite different to those shown for PnirP1 in panel 2B, and in particular, the increase in nirP1 expression at 24 h of 1C is missing in 2B. This should be discussed, or maybe additional experiments carried out to discard the possibility that such increase is an artifact.
8. Line 256: The text says that total amount of amino acids was significantly higher at 12 and 24 h, but the asterisk in figure 4C is only shown for 24 h. This should be corrected. Also, panels A and B in figure 4 should show the units for the Y axis (as in panel C).
9. Legend of figure 4 states that 2 biological replicates were used for the experiments. This is surprising, since it is widely accepted that a minimum of 3 biological replicates should be used in all experiments to get sound conclusions (some companies performing metabolomic determinations suggest up to 5 replicates to ensure reliability). The same comment applies to results shown in figure 5 (also with 2 biological replicates), although in that case it would be acceptable since they are qualitative, and not quantitative.
10. While reading the manuscript section showing that NirP1 interacts with ferredoxin-nitrite reductase, I was expecting the authors would perform a series of enzymatic assays to check

whether NirP1 binding inhibits the nitrite reductase activity, since it seems the next step in the physiological characterization of this process. I am fully aware these are not trivial experiments, but I think these experiments are missing, since they would nicely complement the rest of the information shown in this paper (in particular, to the possible cause that triggers the excretion of nitrite, as described in the next section: inhibition of NiR).

11. Lines 347-349: This sentence states that the results demonstrated, among other things, the inhibition of NiR by NirP1. While I agree in that it is probably the case, the paper shows (for the reasons described above) no results addressing the inhibition of NiR by NirP1. Hence that statement should be deleted, unless additional experiments are carried out to demonstrate the NiR inhibition.

12. Line 350: Only 5% of the assimilated nitrate was excreted as nitrite into the medium in the first 2 h. I wonder if this is physiologically relevant, in light of the comparison to *Prochlorococcus* and *Synechococcus* 7942 used by the authors. Furthermore, there are previous studies in *Synechocystis* 6803 describing an excretion of about 60 % of the reduced nitrate under mixotrophic conditions (Effect of Glucose Utilization on Nitrite Excretion by the Unicellular Cyanobacterium *Synechocystis* sp. Strain PCC 6803 Reyes et al, 1993, *Appl Environ Microbiol*). While these are different conditions, they show that the same organism is capable of excreting more than 50 % of the reduced nitrate. This leads to the following question: if NiR is effectively inhibited by NirP1, why there is such a minor release of nitrite to the medium? Maybe the cells are accumulating most of the nitrite in wait of a return to high C conditions, to resume normal N assimilation? This might point out to a mechanism responsible for deciding whether nitrite is released to the media or accumulated into the cell, depending on the physiological conditions; if the latter case is true, as suggested by the authors, there should be a way to avoid nitrite poisoning in the cell. I think these ideas could be addressed in the discussion, and seem worthy to continue the research in future studies

13. Line 378: "NirP1 targets one of the key enzymes of nitrogen assimilation". Did the authors explore the possibility that there are more NirP1 targets? The results shown in the paper are pretty clear, but additional biological replicates might provide relevant information.

14. Lines 383-391: the modeling of the interactions between NiR/NirP1 and NiR/Fd1 provides a credible explanation for the probable inhibition mechanism of NiR by NirP1. Although I understand these experiments are out of the scope of this paper, this begs for further exploration by enzymatical NiR assays, using NirP1/Fd1 in different ratios to check for the NirP1 inhibitory effect.

15. Line 400-402: The second transcriptional factor possibly involved in the NirP1 regulation might have an enhancing role, on top of NtcA: maybe the small nitrite excretion when NirP1 is expressed might be due to a small inhibition of NiR in the conditions so far tested; so that an NirP1 expression increased by the second transcriptional factor (depending on conditions different from low C) might be decisive to produce a significant nitrite excretion. This is pure speculation in lack of additional evidence, but it is an explanation for the surprisingly low nitrite excretion observed in these studies.

16. Lines 416-435: The hypothesis of nitrite release as a beneficial trait for the microbiome (as proposed in *Prochlorococcus*) is very interesting. The ecological information on *Synechocystis* is rather scarce (in sharp contrast to *Prochlorococcus*), but it might be worth to check for microbial interactions in this organism, to strengthen our understanding of the physiology of this organism.

Minor formal corrections:

Line 53: Remove "(C)", since the abbreviation for carbon is already shown in line 49.

Line 233: Change to "... because the non induced mutant in the absence of added Cu₂SO₄ showed the same pigmentation as the wild type".

Line 439: "maintained BG11 or copper-free BG11" should be changed to "maintained in BG11 or copper-free BG11 medium".

Legend of Figure 1: "(D)" should be in bold and the parentheses mark should be deleted.

Supplemental Dataset 1: the name of the cyanobacterial strain should be shown in one of the columns, for clarity.

Line 469: Manufacturer of ANTI-FLAG antisera is not indicated.

Line: 473: The methods for luxAB genes integration in the genome should be explained in more detail, or a reference included.

Line 479: The volume of cells used for centrifugation is not indicated.

Lines 493-495: The ratio of protease/protein ratios is indicated, but it would be best to include the approximate amount of protein utilized in the digestions.

Line 558: "corrected by signals obtained from the wild type": this should be explained with more detail.

Reviewer #3 (Remarks to the Author):

In the manuscript, No. NCOMMS-23-37089, entitled Nitrite excretion by cyanobacteria is controlled by the small protein NirP1, the authors clearly demonstrated the novel protein, NirP1, directly associated with a nitrite reductase in the Synechocystis and inhibited the activity. The results were very clear and all the experimental procedures to interpret the conclusion were very reasonable. I, as a reviewer, just point out very minor points.

1. In P 3, L54, ammonia (NH₄), nitrite (NO₂), nitrate (NO₃) should be ammonium (NH₄⁺), nitrite (NO₂⁻), nitrate (NO₃⁻).

2. In Fig. 2, the authors showed a tandem repeat sequence, and a putative NtcA-binding site might be involved in regulating the nirP1 expression. And the authors pointed out that at least two regulators might be involved. It is a significant result. The authors compared the primary sequences of the homologs of NirP1 in Fig. 1B. However, the authors did not mention anything about the similarity of the promoter sequences from the homologous genes in other cyanobacteria. If this information is included, the quality of the manuscript should increase.

3. In Fig. 7, the authors summarized the function of NirP1 in the model cyanobacterium. The expression of NirP1 might be regulated with C and N availability. This model only includes the function of N, but not C. If the authors modify or redraw it to include both C and N, it should be nice. The substrate-binding protein of NRT binds both nitrate and nitrite, the indication of nitrite-specific transporter indicated by "?" sounds strange. Also, in this fig, the transporters, NRT and AMY are located in both outer and inner membranes. These should be on the inner membrane of the cells.

Point-to-point replies to the reviews of our manuscript NCOMMS-23-37089 “The small protein NirP1 controls nitrite reductase activity and nitrite excretion in cyanobacteria”

Editorial decision from September 19, 2023

REVIEWER COMMENTS

We are grateful to all three reviewers for their very constructive suggestions!

The herewith submitted version has been thoroughly revised and several additional experiments have been performed. Please find the details in our point-to-point responses below.

Reviewer #1 (Remarks to the Author):

The key results described by Kraus et al., provide evidence of a new mechanism by which nitrite is released from cyanobacteria during growth on nitrate. The authors present an elegant series of experiments that implicate the NirP1 protein in modulating nitrite release by *Synechocystis* by interacting with the nitrite reductase protein NirA. Further, they demonstrate that this NirP1 protein is, itself, likely regulated by NtcA (via repression) and an additional unknown activator. It appears that NirP1 may allow for more precise control over C/N homeostasis within the cyanobacteria that encode this post-translational regulatory protein in their genomes. The hypotheses, methods, statistical analyses, interpretations, and conclusions appear to be robust. I think the work is highly significant given that it is still unclear to the scientific community why some cyanobacteria release nitrite during growth on nitrate (particularly under nutrient limiting conditions). The authors supply some valuable new information on this front that is related to C-limitation in cyanobacteria and manifested via post-translation regulation of nitrite reductase.

Many thanks! These positive and encouraging comments are highly appreciated.

Suggested Improvements:

(1) It would be very useful if Supplemental Dataset 1 was modified to provide information on the organism represented by the sequence record. This would facilitate access to both casual readers and those who want to further explore this phenomenon across cyanobacteria.

Many thanks for this suggestion that makes this dataset more user-friendly!

Supplemental Dataset 1 now includes information on the respective organism as suggested.

(2) Regarding the growth response of the overexpression mutants on nitrate, the data are convincing that there is a growth defect (as well as enhanced nitrite production

during growth on nitrate as presented elsewhere in the manuscript). I was, however, curious about whether or not a growth defect (as well as pigmentation change) would be apparent if the nitrate was replaced entirely by ammonium as the N source. I would hypothesize that the growth and pigmentation defects would be entirely or nearly eliminated given that the nitrate to ammonium reduction pathway would be completely bypassed during growth on ammonium. While this experiment would provide further support for the proposed role of NirP1, the data already provided are quite sufficient. Yet, if the authors already have that data available, I think it would be a nice addition.

Thank you for this suggestion. We performed several experiments in which we washed the strains with medium lacking any N source and then added ammonium as the only N source, as suggested. We recorded a growth curve and performed drop dilution assays and measured absorption spectra with all strains. As expected, the phenotype shown in Figure 3 was abolished in all of these experiments. The phenotype was documented with photos.

We now provide these data in the *SI Appendix* in Supplementary Dataset 4 with the growth curve and the absorption spectra and in the new Supplementary Figure S4.

(3) L263: I think you mean glutamine synthetase rather than GOGAT.

Yes. Correction done as suggested.

(4) Fig. 3: I would appreciate the inclusion of growth rates either overlaid on the figure or provided in a table or within the legend. I see that there is certainly a pronounced lag phase, but it's not immediately clear how different the growth rates are during log-linear growth.

Thanks for this suggestion. The growth rate and doubling time were calculated. The data of the growth curve and the absorption spectra as well as the calculation and values of growth rates and doubling time are given in a Supplementary Dataset 4.

(5) Data availability: Thank you for submitting the proteomics data to a repository. Have the authors also considered submitting the data on nitrate consumption, nitrite production, and OD values to a data repository?

We very much appreciate your suggestion, but unfortunately, we are not aware of any data repository for such data. Therefore, the data on nitrate consumption, nitrite production, and OD values are provided as Excel files in the Supplementary Dataset 7 and uploaded together with this manuscript.

(6) Clarity and context: Overall, the writing is clear and easy to follow. But the authors might want to consult with Nat Comm on which acronyms should be defined. I recommend rephrasing the sentence on lines 326-327 for clarity given its confusing syntax: "However, this concentration was with ~75 μ M much higher for the nirP1-overexpressing strain".

Yes, non-common acronyms are defined. The mentioned sentence has been rephrased as suggested.

Reviewer #2 (Remarks to the Author):

This paper describes a very nice study focused on a previously unknown gene they call nirP1 in *Synechocystis* PCC 6803. The authors used several approaches to investigate the function of this protein: they show that part of a previously observed ncRNA actually encodes a small protein, highly conserved in cyanobacterial genomes (with the exception of *Prochlorococcus* and symbiotic strains as UCYN-A). nirP1 transcription is controlled by NtcA and another transcriptional factor, being expressed under low carbon conditions. Deletion or ectopic expression of this gene lead to clear effects on the growth and color of cultures. The concentration of some nitrogen-related metabolites (in particular, glutamate) was affected in mutants overexpressing nirP1. Co-immunoprecipitation and Western blot experiments showed NirP1 binds to nitrite reductase, suggesting it is a regulator (inhibitor) of this enzyme. Finally, the authors observed that wild type *Synechocystis* 6803 does excrete nitrite under low carbon conditions, while the nirP1 deletion mutant does not, as a consequence of nitrite reductase inhibition. They also demonstrate that the accumulated nitrite is a result from nitrate uptake.

This manuscript contains a substantial amount of work, and the experimental design has been carefully carried out, addressing some of the main questions on the function of this gene. In my opinion it provides a significant advance in the biology of cyanobacterial metabolism and its regulation. Hereafter I suggest some ideas for improvement:

Many thanks for these positive comments. We really appreciate the ideas for improvement!

1. The results shown in the paper demonstrate that NirP1 is a small protein involved in the regulation of nitrite reductase in *Synechocystis* PCC 6803. This results in nitrite excretion (although in uncertain ways, which I will discuss later). Hence I think the title should be based on the concept of nitrite reductase binding/regulation by NirP1, which is the core of the paper, rather than nitrite excretion.

Following this advice, we have changed the title to: “*The nitrite reductase interacting protein NirP1 controls nitrite excretion in cyanobacteria*”

2. Lines 137-144: The authors say that nirP1 is present in genomes from all morphological subsections of cyanobacteria, with the exception of *Prochlorococcus* and UCYN-A. I wonder if they checked the genomes of all available marine strains of *Synechococcus*, and if so, do all of them possess this gene?

Supplementary Dataset 1 has been modified as suggested and now contains the taxonomy IDs and names of the corresponding

organisms. As a result, it is now easier to answer this and similar questions in the future.

Most of the marine *Synechococcus* strains do not have a homologue of *nirP1*. We now mention this in the manuscript: “*Homologous genes were detected in marine species with deviating pigmentation, such as *Acaryochloris marina* and *Halomicronema hongdechloris*, but not in *Prochlorococcus* and not in the majority of marine picoplanktonic *Synechococcus* strains.*”

3. Line 155: The text says that the strongest signal was obtained 1 h after the shift “with somewhat declining intensity at the later time points”. However, the signal corresponding to 24 h is also strong, which does not fit with the description of results. Could the authors provide some explanation for such a high signal after 24 h?

The statement has been corrected to “*A strong signal was obtained 1 h after the shift, with a slightly decreasing intensity at the later time points and increasing again at the 24 h sampling point.*”

We assume that there are two phases of acclimation to low carbon. A first, response triggering the initial strong activation of expression, leading to an initial relieve from carbon limitation, which is followed approximately 24 h later by a second, likely longer-lasting response.

Line 162: The text says “single detected signal” of ca. 12 kDa in Fig. 1D, but there are actually two bands in this figure, which in samples R1 and R2 without NHCO₃ are especially remarkable. I missed an explanation for the presence of this second band (located above) and why it is more intense in the samples without NHCO₃.

Many thanks for this comment. Yes, the explanation of the second band was missing. This section has been modified as follows: “*The calculated molecular mass of ~12 kDa for the prominent signal in the samples from HC-grown cultures was consistent with the sum of the calculated molecular masses of 9.18 kDa for the monomeric NirP1 and 2.86 kDa for the 3xFLAG tag (Fig. 1d). After the shift to LC conditions, a band migrating with a ~1.5 kDa larger molecular mass became more intense, possibly indicating a post-translational modification.*”

5. Lines 184-186: Although Figure 2 shows clearly two mutated versions of the promoter, the text is somehow confusing and the reader might believe there is a single mutated promoter with changes in both the NtcA binding site and in the upstream element; I suggest changing the text for clarity.

Thank you very much for spotting this potentially confusing statement. The sentence has been amended to: “*We used the native P_{nirP1} sequence or promoter variants with mutated, likely relevant nucleotides either in the NtcA binding site ($P_{NtcA-Mut}$) or in the upstream promoter element ($P_{Repeat-Mut}$; Fig. 2a).*”

6. Lines 195-196: Did the authors perform any experiment to assess the identity of this second transcription factor possibly involved in the control of *nirP1*? In any case, it might be interesting to speculate on the possibilities in the discussion.

The second transcription factor involved in the control of *nirP1* is activating transcription and recognizes the motif TTTGT(T/C)AA-N2-TTTGT(T/C)AA, located 43 to 60 nt upstream the transcription start site (Fig. 2A).

At least four different transcription factors are known that are directly involved in the response to variations in Ci levels. These are CmpR, the activator of the ABC-type bicarbonate transporter¹, NdhR, the repressor of high-affinity carbon uptake², RbcR, the regulator of RuBisCO expression³ and cyAbrB2, which was reported as supplementing the functions of NdhR and CmpR in the regulation of several genes related to carbon concentration, such as *sbtA/B*, *ndhF3/ndhD3/cupA* and *cmpABCD*, under low carbon (LC) conditions⁴.

From these, RbcR is a positively acting transcription factor that recognizes the motif ATTA(G/A)-N5-(C/T)TAAT³ which shares the richness in AT nucleotides with the here described element. However, the motif found in the here investigated *nirP1* promoters differed. Moreover, the manipulation of RbcR expression revealed in microarray analyses no impact on *nirP1* expression³, although this might not have been possible to see if NtcA was present in that strain, effectively continuing to block transcription.

CmpR is an activator as well and recognizes an AT-rich motif, too, described as TTA-N7/8-TAA¹. It is possible to superimpose this consensus on the motif in the *nirP1* promoter, but the actual motif is longer than this 13- to 14-meric recognition sequence.

Therefore, only experimental analysis can reveal the nature of this second transcription factor. In this work, we only performed promoter assays to validate the binding motifs of transcription factors controlling *nirP1* during the shift to LC conditions and N starvation. Since the characterization of the very factors binding these motifs was not the main topic of this work, we did not follow it further. Nevertheless, the nature of the second transcription factor is an interesting topic for further work.

7. The results shown in panel 1C are quite different to those shown for P_{nirP1} in panel 2B, and in particular, the increase in *nirP1* expression at 24 h of 1C is missing in 2B. This should be discussed, or maybe additional experiments carried out to discard the possibility that such increase is an artifact.

Thanks, for this comment. Both Fig. 1c and Fig. 2b show similarities and differences. In Fig. 2b, there is a slight increase noticeable for the NtcA-mutant starting at 12 h and after 6 h for P_{nirP1}. The increase 24 h

in the promotor assay shown in Fig. 2b is not as strong as in the northern blot shown in Fig. 1c. The reason for this difference is, that both figures show the results of completely different experiments.

In Fig. 2b we show the promotor activity of *nirP1*, where we fused the promotor sequence of *nirP1* to the mRNA coding for luciferase. Activation of P_{nirP1} in shifts to LC led to the transcription of the luciferase mRNA. After translation of this mRNA, finally, the light signal measured in the promotor assay in Fig. 2b is the result of the degradation of decanal by the enzyme luciferase.

In contrast, the northern blot in Fig. 1d shows the inducibility of *nirP1* after shifts to LC conditions as well, but at transcript level. The steady-state level of a transcript depends on the rate of transcription and the rate of degradation, determining the stability of the transcript and the resulting lifetime.

Compared to the luciferase mRNA, the *nirP1* mRNA is a completely different transcript from the *nirP1* mRNA. These transcripts differ in composition, length and secondary structures. Therefore, different properties in terms of stability, lifetime, rate of production and degradation can be expected.

Therefore, the increased *nirP1* mRNA levels visible after 24 h at LC in Fig. 1C likely resulted from a post-transcriptional mechanism stabilizing the mRNA.

We have inserted 1.5 sentences to refer to this possibility: “*In strains P_{nirP1} and $P_{NtcA-Mut}$, transcription was activated 1 h after a shift to medium lacking C_i , followed by a decrease and a light increase again at the later time points (Fig. 2b, Supplementary Fig. S3). The increase at 24 h was not as pronounced as in the northern blot (Fig. 1c), pointing at the possible involvement of additional post-transcriptional regulation.*”

8. Line 256: The text says that total amount of amino acids was significantly higher at 12 and 24 h, but the asterisk in figure 4C is only shown for 24 h. This should be corrected. Also, panels A and B in figure 4 should show the units for the Y axis (as in panel C).

Thank you for these recommendations. All points have been corrected in Fig. 4 and the axis labels are now consistent. The text has been changed and adapted to: “*The total amount of soluble amino acids was significantly higher in $NirP1oex$ at 24 h after transfer to LC than in the other two strains (Fig. 4a, full data in Supplemental Dataset 5).*”

9. Legend of figure 4 states that 2 biological replicates were used for the experiments. This is surprising, since it is widely accepted that a minimum of 3 biological replicates should be used in all experiments to get sound conclusions (some companies performing metabolomic determinations suggest up to 5 replicates to ensure reliability). The same comment applies to results shown in figure 5 (also with 2

biological replicates), although in that case it would be acceptable since they are qualitative, and not quantitative.

Thanks for this input. We have repeated the experiment previously shown in Fig. 4 and Supplemental Dataset 5. The new data are provided with this manuscript in Fig. 4c and the repeated measurements in Supplemental Dataset 6. As suggested, we used 3 biological replicates for each strain in the new experiment. The trends in the new dataset (Supplemental Dataset 6) are the same as in the previous analysis. The data of the first dataset were thus confirmed in independent measurements and with more involved replicates.

Additionally, we were now also able to measure the 2-OG levels. These levels were significantly higher in strain NirP1oex than in the other strains. This is consistent with the pigmentation phenotype upon the ectopic overexpression of *nirP1* and indicates an impaired regulation of the GOGAT cycle and reduced N assimilation. This is also visible in the glutamine and glutamate concentrations in Dataset 5, which is also consistent with the older results. Thus, an additional copy of the *nirP1* gene leads to a higher accumulation of 2-OG. These data are shown in Fig. 4c.

To summarize, the new data provided in Dataset 6 are an extension of the data shown in Fig. 4 and Supplemental Dataset 5. We refer to the new measurements in the legend of Fig. 4.

10. While reading the manuscript section showing that NirP1 interacts with ferredoxin-nitrite reductase, I was expecting the authors would perform a series of enzymatic assays to check whether NirP1 binding inhibits the nitrite reductase activity, since it seems the next step in the physiological characterization of this process. I am fully aware these are not trivial experiments, but I think these experiments are missing, since they would nicely complement the rest of the information shown in this paper (in particular, to the possible cause that triggers the excretion of nitrite, as described in the next section: inhibition of NiR).

Thank you very much! The characterization of NirP1 in enzymatic assays is indeed of great interest to us. We performed enzymatic assays with cell cultures and with purified enzymes.

During measurements of nitrate reductase activity assays using cell suspensions we see differences in nitrite concentrations in the supernatant of wild type, deletion mutant and overexpressor strains. This is consistent with the results shown Fig 6., that the strains expressing *nirP1* start to secrete nitrite to the supernatant. But we couldn't observe different NiR activities in these strains directly. The underlying reason seems to be that the complex of NirP1 and NiR is redox-sensitive. This fits to the presence of cysteine residues in likely critical positions of the molecule and to the transcriptional regulation of *nirP1*, as it is upregulated under LC conditions and downregulated under nitrogen starvation and is thus dependent on the C/N balance.

By using NaDT as electron donor in the *in-vitro* assays we changed the redox potential of the cells. The difficulty is that, on the one hand, NirP1 appears to release NiR under reducing conditions yielding active NiR, but on the other hand, we cannot simply omit the electron donor because the electrons are required for the reduction of nitrite.

It is possible that very sensitive fine tuning of assay components and conditions will finally help to make the enzymatic assays productive. We are working on this question and this is still ongoing. No change in manuscript.

11. Lines 347-349: This sentence states that the results demonstrated, among other things, the inhibition of NiR by NirP1. While I agree in that it is probably the case, the paper shows (for the reasons described above) no results addressing the inhibition of NiR by NirP1. Hence that statement should be deleted, unless additional experiments are carried out to demonstrate the NiR inhibition.

Yes, that is correct. Since we have not yet been able to demonstrate the inhibition of NiR by NirP1, this sentence has been adjusted as suggested to: *“These results demonstrated that the accumulation of nitrite in the medium of the wild type and overexpression strains resulted from nitrate uptake, the subsequent reduction to nitrite by NR and the alteration of NiR activity, with the involvement of NirP1.”*

12. Line 350: Only 5% of the assimilated nitrate was excreted as nitrite into the medium in the first 2 h. I wonder if this is physiologically relevant, in light of the comparison to *Prochlorococcus* and *Synechococcus* 7942 used by the authors. Furthermore, there are previous studies in *Synechocystis* 6803 describing an excretion of about 60 % of the reduced nitrate under mixotrophic conditions (Effect of Glucose Utilization on Nitrite Excretion by the Unicellular Cyanobacterium *Synechocystis* sp. Strain PCC 6803 Reyes et al, 1993, *Appl Environ Microbiol*). While these are different conditions, they show that the same organism is capable of excreting more than 50 % of the reduced nitrate. This leads to the following question: if NiR is effectively inhibited by NirP1, why there is such a minor release of nitrite to the medium? Maybe the cells are accumulating most of the nitrite in wait of a return to high C conditions, to resume normal N assimilation? This might point out to a mechanism responsible for deciding whether nitrite is released to the media or accumulated into the cell, depending on the physiological conditions; if the latter case is true, as suggested by the authors, there should be a way to avoid nitrite poisoning in the cell. I think these ideas could be addressed in the discussion, and seem worthy to continue the research in future studies.

Many thanks for pointing at this interesting publication! Moreover, the idea that the remaining nitrite could be stored somewhere in the cell to avoid nitrite poisoning is intriguing. We have indeed circumstantial evidence for such a nitrite storage, possibly in the periplasm. When we washed the NirP1 overexpressor strain and transferred it to nitrate-containing media we observed a very rapid initial increase in the

nitrite concentration in the medium (see time point 0 in Fig. 6a), which then was followed only by slower increase.

Following this advice, we have transferred the statement previously beginning in line 350 from the Results section to the Discussion where we now continue by briefly discussing these considerations. This section now constitutes the second half of the penultimate paragraph of the Discussion.

13. Line 378: “NirP1 targets one of the key enzymes of nitrogen assimilation”. Did the authors explore the possibility that there are more NirP1 targets? The results shown in the paper are pretty clear, but additional biological replicates might provide relevant information.

Yes, this is correct! In this work we show that NiR is a real interaction partner of NirP1 and focused on this result. The reason for this was, that we noticed the prominent band in the protein gel co-eluting with NirP1 in pulldown experiments. This was observed in protein gels and also in western blots detecting NirP1 migration at a higher molecular mass compared to the mass of monomeric protein.

Since the NiR was surprisingly co-eluted only after 8 washing steps, the interaction of the two proteins is very strong and NiR seems to be the main (or one of the main) interaction partners. We are aware that there may be other interaction partners of NirP1. Their identity should clarify the regulation by NirP1 further. Such experiments are planned for the future.

14. Lines 383-391: the modeling of the interactions between NiR/NirP1 and NiR/Fd1 provides a credible explanation for the probable inhibition mechanism of NiR by NirP1. Although I understand these experiments are out of the scope of this paper, this begs for further exploration by enzymatical NiR assays, using NirP1/Fd1 in different ratios to check for the NirP1 inhibitory effect.

Thank you very much for this suggestion. Yes, the inclusion of Fd1 for the characterization of NirP1 makes totally sense. This is also the plan for further experiments to follow the key hypothesis of the current work. Here, we wanted to highlight the new regulator involved in the regulation of the nitrogen assimilation and its role on nitrite excretion.

15. Line 400-402: The second transcriptional factor possibly involved in the NirP1 regulation might have an enhancing role, on top of NtcA: maybe the small nitrite excretion when NirP1 is expressed might be due to a small inhibition of NiR in the conditions so far tested; so that an NirP1 expression increased by the second transcriptional factor (depending on conditions different from low C) might be decisive to produce a significant nitrite excretion. This is pure speculation in lack of additional evidence, but it is an explanation for the surprisingly low nitrite excretion observed in these studies.

Thanks for this comment. At the moment it is hard to comment on the low nitrite excretion as this is the starting point of this studies. We can only speculate on this. We noticed the difference and will therefore investigate this more closely in the future.

16. Lines 416-435: The hypothesis of nitrite release as a beneficial trait for the microbiome (as proposed in *Prochlorococcus*) is very interesting. The ecological information on *Synechocystis* is rather scarce (in sharp contrast to *Prochlorococcus*), but it might be worth to check for microbial interactions in this organism, to strengthen our understanding of the physiology of this organism.

Yes, that is correct. There is a lack of information on *Synechocystis* microbial interactions, which should be studied in more detail in the future. The process of cross-feeding shown for *Prochlorococcus* could also be relevant for *Synechocystis*, and it would make sense that this would happen in a more regulated way.

Minor formal corrections:

Line 53: Remove "(C)", since the abbreviation for carbon is already shown in line 49.

Thank you very much for this suggestion! We have used two different abbreviations, because carbon mentioned in line 49 refers to carbon in general (C) and later in line 53 the abbreviation (Ci) refers to inorganic carbon. *"Some cyanobacteria, such as marine Prochlorococcus and Synechococcus, are of paramount importance as primary producers and for the global biogeochemical carbon (C) and nitrogen (N) cycles. The two most relevant nutrients for cyanobacteria are inorganic carbon (Ci) and N."*

Line 233: Change to "... because the non induced mutant in the absence of added Cu₂SO₄ showed the same pigmentation as the wild type".

Yes, the wording was confusing. The sentence has been corrected as suggested to: *"This effect was strictly dependent on the nirP1 expression level because the non-induced mutant showed the same pigmentation as the wild type in the absence of added Cu₂SO₄ (Supplementary Fig. S3a)."*

Line 439: "maintained BG11 or copper-free BG11" should be changed to "maintained in BG11 or copper-free BG11 medium".

Thanks, this has been corrected as suggested to: *"Strains were maintained in BG11 or in copper-free BG11⁵ supplemented with 20 mM TES pH 7.5 under continuous white light of 50 μmol photons m⁻² s⁻¹ at 30 °C."*

Legend of Figure 1: "(D)" should be in bold and the parentheses mark should be deleted.

Thank you, correction done as suggested. The legend of Figure 1 has been adapted to the general figure guidelines of *Nature Communications*. This was also done for all other figures.

Supplemental Dataset 1: the name of the cyanobacterial strain should be shown in one of the columns, for clarity.

Yes, the names of all cyanobacterial strains have been included now.

Line 469: Manufacturer of ANTI-FLAG antisera is not indicated.

Yes, that's right and was a mistake. Manufacturer was added: *"The expression of nirP1 was checked by Western hybridization using ANTI-FLAG antisera (Sigma)."*

Line: 473: The methods for luxAB genes integration in the genome should be explained in more detail, or a reference included.

Yes, we did not go into detail because the generation of reporter strains and the luciferase assay were described in a previous publication. This publication has already been mentioned as a reference in this work. *"The selection of the reporter strains and bioluminescence measurements were performed as previously described⁶."*

Line 479: The volume of cells used for centrifugation is not indicated.

Yes, this information was missing and has been added: *"Cells (50 mL) were harvested by centrifugation (5,000 × g, 4 °C, 10 min) after 24 h of NirP1-3xFLAG overexpression, resuspended in FLAG buffer."*

Lines 493-495: The ratio of protease/protein ratios is indicated, but it would be best to include the approximate amount of protein utilized in the digestions.

Thank you very much for this suggestion. The section has been changed and the amount of protein has been added: *"For further analysis, samples were resolubilized in denaturation buffer (6 M urea, 2 M thiourea in 100 mM Tris/HCl; pH 8.0) and the protein concentration was measured again by Bradford assay. Subsequently, 0.1 mg protein was separated per sample and adjusted to a final protein concentration of 1 μg/μL."*

Line 558: "corrected by signals obtained from the wild type": this should be explained with more detail.

Many thanks! This was not intentional. This section has been modified and corrected. More details of the method are described in SI Appendix in Supplementary methods. *"Authentic standard substances (Merck) at various concentrations were used for calibration, and peak areas were normalized to signals of the internal standard (carnitine). The data were further normalized to the OD₇₅₀ and volume measured for each sample."*

Reviewer #3 (Remarks to the Author):

In the manuscript, No. NCOMMS-23-37089, entitled Nitrite excretion by cyanobacteria is controlled by the small protein NirP1, the authors clearly demonstrated the novel protein, NirP1, directly associated with a nitrite reductase in the Synechocystis and inhibited the activity. The results were very clear and all the experimental procedures to interpret the conclusion were very reasonable.

We are happy to hear your positive feedback. Thank you very much!

I, as a reviewer, just point out very minor points.

1. In P 3, L54, ammonia (NH₄), nitrite (NO₂), nitrate (NO₃) should be ammonium (NH₄⁺), nitrite (NO₂⁻), nitrate (NO₃⁻).

Thank you very much. That was a mistake. The correction has been made to: *“Most cyanobacteria assimilate combined inorganic N forms, such as ammonia (NH₄⁺), nitrate (NO₃⁻), nitrite (NO₂⁻) and urea; in addition, diazotrophic species can utilize N₂ gas.”*

2. In Fig. 2, the authors showed a tandem repeat sequence, and a putative NtcA-binding site might be involved in regulating the nirP1 expression. And the authors pointed out that at least two regulators might be involved. It is a significant result. The authors compared the primary sequences of the homologs of NirP1 in Fig. 1B. However, the authors did not mention anything about the similarity of the promoter sequences from the homologous genes in other cyanobacteria. If this information is included, the quality of the manuscript should increase.

Thank you very much for this great suggestion. Following this suggestion, a sequence alignment of 456 promoter sequences of the 485 nirP1 homologs from Dataset 1 was performed. For 29 homologs upstream sequence information was missing.

The results are provided in a new file together with all relevant details as Dataset 2. In addition, the conservation of nucleotides of the motifs that are shown in Fig. 2, as the NtcA binding motif and the tandem repeat, was visualized with MEME and is now shown in Supplementary Fig. S2.

3. In Fig. 7, the authors summarized the function of NirP1 in the model cyanobacterium. The expression of NirP1 might be regulated with C and N availability. This model only includes the function of N, but not C. If the authors modify or redraw it to include both C and N, it should be nice. The substrate-binding protein of NRT binds both nitrate and nitrite, the indication of nitrite-specific transporter indicated by “?” sounds strange. Also, in this fig, the transporters, NRT and AMY are located in both outer and inner membranes. These should be on the inner membrane of the cells.

Thank you for these excellent recommendations. Figure 7 has been improved as suggested.

The second transcription factor responsible for activation in LC conditions is still unknown. Nevertheless, we have shown both

transcription factors in Figure 7. Therefore, the transcription factor responsible for the activation of *nirp1* is referred to as TF. The activation of *nirP1* by TF is indicated by an arrow.

References in this letter

1. Nishimura, T. *et al.* Mechanism of low CO₂-induced activation of the *cmp* bicarbonate transporter operon by a LysR family protein in the cyanobacterium *Synechococcus elongatus* strain PCC 7942. *Mol Microbiol* **68**, 98–109 (2008).
2. Klähn, S. *et al.* Integrated transcriptomic and metabolomic characterization of the low-carbon response using an *ndhR* mutant of *Synechocystis* sp. PCC 6803. *Plant Physiol.* **169**, 1540 (2015).
3. Bolay, P. *et al.* The transcriptional regulator RbcR controls ribulose-1,5-bisphosphate carboxylase/oxygenase (RuBisCO) genes in the cyanobacterium *Synechocystis* sp. PCC 6803. *New Phytol* **235**, 432–445 (2022).
4. Orf, I. *et al.* CyAbrB2 contributes to the transcriptional regulation of low CO₂ acclimation in *Synechocystis* sp. PCC 6803. *Plant Cell Physiol.* **57**, 2232–2243 (2016).
5. Rippka, R., Deruelles, J., Waterbury, J. B., Herdman, M. & Stanier, R. Y. Generic assignments, strain histories and properties of pure cultures of cyanobacteria. *Microbiology* **111**, 1–61 (1979).
6. Klähn, S. *et al.* Alkane biosynthesis genes in cyanobacteria and their transcriptional organization. *Front. Bioeng. Biotechnol.* **2**, 24 (2014).

Reviewer #1 (Remarks to the Author):

The authors have satisfactorily addressed all of my comments/suggestions in their revision.

Reviewer #2 (Remarks to the Author):

Most of the comments and suggestions I made about for the previous version of this manuscript have been addressed by the authors; furthermore, I am pleased to see they have made new experiments, providing confirmation of results (with additional replicates) and also new information regarding the concentrations of 2-oxoglutarate (as explained in page 7 of the rebuttal letter).

I have only three minor suggestions:

- The new title ("The nitrite reductase interacting protein NirP1 controls nitrite excretion in cyanobacteria") is still stressing the control of nitrite excretion as the main role of NirP1. However, for the reasons I explained in my comments to the initial version of this paper, I keep on thinking the title should focus on the control of NiR by NirP1, rather than on nitrite excretion (especially, given the low amount of nitrite being excreted under their experimental conditions; i.e., 5 % of the assimilated nitrate). But of course the authors are free to keep the current title.
- The revised version includes in fig 2C data on the concentrations of 2-oxoglutarate; however, I think the method used to determine them has not been included in Materials and Methods. Or at least, I could see no change in that section to describe the 2-oxoglutarate determination.
- The legend of Fig. 4 contains two conflicting sentences regarding the number of biological replicates for metabolites: one says "Three biological replicates of all strains were used and averaged". The other, "Two biological replicates of all strains were used for all metabolites (except 2-oxoglutarate) and averaged". One of them should be removed, leaving only one with the correct information.

Point-to-point replies to the reviews of our manuscript NCOMMS-23-37089A
“The nitrite reductase interacting protein NirP1 controls nitrite excretion in cyanobacteria”, now changed to

“Protein NirP1 regulates nitrite reductase and nitrite excretion in cyanobacteria”

Editorial decision from February 02, 2024

REVIEWER COMMENTS

Reviewer #1 (Remarks to the Author):

The authors have satisfactorily addressed all of my comments/suggestions in their revision.

Many thanks! Your comment is highly appreciated.

Reviewer #2 (Remarks to the Author):

Most of the comments and suggestions I made about for the previous version of this manuscript have been addressed by the authors; furthermore, I am pleased to see they have made new experiments, providing confirmation of results (with additional replicates) and also new information regarding the concentrations of 2-oxoglutarate (as explained in page 7 of the rebuttal letter).

Many thanks for these positive comments. We have really appreciated the suggestions for improvement!

I have only three minor suggestions:

- The new title ("The nitrite reductase interacting protein NirP1 controls nitrite excretion in cyanobacteria") is still stressing the control of nitrite excretion as the main role of NirP1. However, for the reasons I explained in my comments to the initial version of this paper, I keep on thinking the title should focus on the control of NiR by NirP1, rather than on nitrite excretion (especially, given the low amount of nitrite being excreted under their experimental conditions; i.e., 5 % of the assimilated nitrate). But of course the authors are free to keep the current title.

We have changed the title to: “Protein NirP1 regulates nitrite reductase and nitrite excretion in cyanobacteria”

This variant follows the advice to focus on the control of NiR by NirP1, but covers also the second most relevant aspect of our study.

Hence, it will make both aspects directly accessible to the readers.

- The revised version includes in fig 2C data on the concentrations of 2-oxoglutarate; however, I think the method used to determine them has not been included in Materials and Methods. Or at least, I could see no change in that section to describe the 2-oxoglutarate determination.

2-OG has been measured using the same protocol as for all other metabolites. We used the LC-MS/MS device and MRM (multiple reaction monitoring) in the negative method. Therefore, the section “Extraction and Metabolite Analysis” includes the methodology for measuring 2-OG as well as all other metabolites. No change.

- The legend of Fig. 4 contains two conflicting sentences regarding the number of biological replicates for metabolites: one says "Three biological replicates of all strains were used and averaged". The other, "Two biological replicates of all strains were used for all metabolites (except 2-oxoglutarate) and averaged". One of them should be removed, leaving only one with the correct information.

Many thanks, this was indeed a confusing description. The legend of Fig. 4 has been revised accordingly.